# Observations of positive sea surface temperature trends in the steadily shrinking Dead Sea

Pavel Kishcha[1], Rachel T. Pinker[2], Isaac Gertman[3], Boris Starobinets[1], Pinhas Alpert[1]

[1]Department of Geophysics, Tel Aviv University, Tel-Aviv, 69978, Israel
[2]Department of Atmospheric and Oceanic Science, University of Maryland, College Park, MD 20742, USA
[3]Israel Oceanographic and Limnological Research, National Institute of Oceanography, Haifa, 31080, Israel

*Correspondence to*: Pavel Kishcha (pavel@cyclone.tau.ac.il)

**Abstract.** Increasing warming of steadily shrinking Dead Sea surface water compensates for surface water cooling (due to increasing evaporation) and even causes observed positive Dead Sea surface temperature trends. This warming is caused by two factors: increasing daytime heat flow from land to sea (as a result of the steady shrinking) and regional atmospheric warming. Using observations from Moderate Resolution Imaging Spectroradiometer (MODIS), positive trends were detected in both daytime and nighttime Dead Sea surface temperature (SST) over the period of 2000 – 2016. These positive SST trends were observed in the absence of positive trends in surface solar radiation, measured by the Dead Sea buoy pyranometer. We also show that long-term changes in water mixing in the uppermost layer of the Dead Sea under strong winds could not explain the observed SST trends. There is a positive feedback loop between the positive SST trends and the shrinking of the Dead Sea, which contributes to the accelerating decrease in Dead Sea water levels during the period under study. Satellite-based SST measurements showed that maximal SST trends of over 0.8 $^{o}$C decade$^{-1}$ were observed over the north-west and southern sides of the Dead Sea, where shrinking of the Dead Sea water area was pronounced. No noticeable SST trends were observed over the eastern side of the lake, where shrinking of the Dead Sea water area was insignificant. This finding demonstrates correspondence between the positive SST trends and the shrinking of the Dead Sea indicating a causal link between them. There are two opposite processes taking place in the Dead Sea: sea surface warming and cooling. On the one hand, the positive feedback loop leading to sea surface warming every year accompanied by long-term increase in SST; on the other hand, the measured acceleration of the Dead Sea water-level drop suggests a long-term increase in Dead Sea evaporation accompanied by a long-term decrease in SST. During the period under investigation, the total result of these two opposite processes is the statistically significant positive sea surface temperature trends in both daytime (0.6 $^{o}$C decade$^{-1}$) and nighttime (0.4 $^{o}$C decade$^{-1}$), observed by the MODIS instrument. Our findings of the existence of a positive feedback loop between the positive SST trends and the shrinking of the Dead Sea imply the following significant point: any meteorological, hydrological or geophysical process causing the steady shrinking of the Dead Sea will contribute to positive trends in SST. Our results shed light on continuing hazards to the Dead Sea.

# 1 Introduction

The coastal area of the hypersaline terminal lake of the Dead Sea is a unique area of dry land of the lowest elevation on Earth (-420 m a.s.l.). Solar radiation heats this dry coastal area in the daytime and creates a temperature gradient between the uppermost levels of the land and those of the sea. The Dead Sea has been drying up over the last two decades: the water level dropped at the rate of approximately 1 m year$^{-1}$ (Lensky et al., 2005). The Dead Sea drying up is due to the lack of water inflow from the Jordan River; a decreasing tendency in rainfall over the last 40 years (Ziv et al., 2015); and evaporation (Lensky et al, 2005, 2018, Metzger et al., 2018, Alpert et al., 1997, Shafir and Alpert, 2011, AL-Khlaifat, 2018). The Dead Sea drying up has led to the shrinking in the Dead Sea water area. Based on satellite imagery from 1972 to 2013, El-Hallaq and Habboub (2014) estimated that the Dead Sea water area shrank on average at the rate of ~2.9 km$^2$ year$^{-1}$.

Knowledge of the Dead Sea thermal structure has been gained from measured water temperature vertical profiles of over the past 40 years (Gertman and Hecht, 2002; Hecht and Gertman, 2003; Kishcha et al., 2017; Nehorai et al., 2009; Stanhill, 1990). Using regular buoy measurements of Dead Sea water temperature at the depth of 1 m during the ten year period from 1992 to 2002, Hecht and Gertman (2003) detected an increasing statistically significant trend of 0.06 $^o$C year$^{-1}$.

Dead Sea surface temperature (SST), which is the main point of our study, is one of the causal factors of water evaporation which affects the Dead Sea water level (Lensky et al., 2018). There are only a few studies on the Dead Sea SST (Nehorai et al., 2009, 2013; Stanhill, 1990). The above mentioned studies dealt with diurnal, seasonal and interannual variations in SST. O'Reilly et al. (2015) discussed a statistically significant positive trend of 0.34 $^o$C per decade in the nighttime surface water temperature in approximately 300 lakes around the world including the Dead Sea (characterized by the statistically significant positive trend of 0.63 $^o$C per decade). This was achieved by using both satellite and in situ measurements in the summer season (from July to September) during the 25-year period from 1985 – 2009. They consider that the increase was associated with the interaction among different climatic factors such as increasing surface solar radiation as a result of decreasing cloud cover and increasing air temperature (O'Reilly et al., 2015). To our knowledge, long-term interannual sea surface temperature changes in both daytime and nighttime periods, taking into account all the months of the year, have not been discussed in previous publications.

Our study aims at investigating long-term trends in the Dead Sea SST using the 17-year MODIS period of records (2000 – 2016). This study was carried out on skin surface temperature over land and sea using MODIS data on board the NASA Terra satellite. We found statistically significant positive trends in Dead Sea SST in the absence of positive trends in surface solar radiation which raise questions about the factors contributing to Dead Sea water heating.

# 2 Method

For the remotely sensed monthly mean temperatures of the Dead Sea, we used Collection-6 (C6) of the MODIS Land Surface Temperature (LST) Product: MOD11C3 Level 3 (Wan, 2014). Wan (2014) showed that LST data from Collection-6 are more accurate than those from the previous Collection 5: the mean C6 LST error is within $\pm$ 0.6 $^o$C which is lower than

the mean C5 LST error of ± 2 °C. The gridded MOD11C3 data are available at 5 km spatial resolution, two times per day: in the daytime at approximately 10:30 LT and in the nighttime at ~21:30 LT. To study sea surface temperature (SST) trends we used only pixels which covered the Dead Sea (Fig. 1, blue boxes), while all others have been eliminated from the analysis, to avoid thermal contamination. In addition to SST trends, we analyzed similar long-term trends of the skin surface temperature over the land area in the vicinity of the Dead Sea (Fig. 1, red boxes).

To obtain long-term trends of the Dead Sea SST, the above-mentioned MOD11C3 Level 3 monthly data averaged over the Dead Sea (Fig. 1, blue boxes) were deseasonalized by removing 17-year averages from any given month. A similar approach was used in order to obtain long-term trends of the skin surface temperature over the land area (Fig. 1, red boxes). The slope of a linear fit was used to determine Dead Sea surface temperature trends as well as those of skin surface temperature over the land, during the 17-year period under investigation (2000 – 2016). To estimate the significance level (p) value of surface temperature trends, normally distributed residuals of the linear fit were used in a t test (Shapiro and Wilk, 1965; Razali and Wah, 2011). The obtained p values less than 0.05 correspond to statistically significant surface temperature trends at the 95% confidence level.

To study the effect of climatic factors on long-term trends in the Dead Sea SST, we used available pyranometer measurements of surface solar radiation together with measurements of near-surface wind speed from a hydrometeorological buoy, anchored in the Dead Sea (Fig. 1). The measured surface solar radiation (SR) was represented by monthly data of daily average SR and those of daily maximum SR during the 9-year period from 2005 to 2013. The measured wind speed was represented by monthly data of daily averaged near-surface wind speed, during the 10-year period from 2005 - 2014. We also analyzed monthly data of daily averaged near surface wind speed from two other meteorological stations, located in the vicinity of the Dead Sea: Sdom (31.03N, 35.39E) (the 13-year period from 2004 – 2016) and Ein-Gedi-SPA (31.42N; 35.38E) (the 10-year period from 2007 – 2016).

To estimate long-term trends of above mentioned climatic factors, the same approach was used as for surface temperature trends. In addition, we analyzed monthly data of Dead Sea water levels based on available measurements from 1992 until the present. Taking into account that long-term changes in Dead Sea water levels reflect changes in Dead Sea evaporation, the measurements of Dead Sea water levels were used for the analysis of a possible contribution of long-term changes in evaporation to long-term trends in the Dead Sea SST.

## 3 Results

### 3.1 Trends in Dead Sea SST and surface solar radiation

MODIS satellite data of skin surface temperature allowed us to estimate long-term trends in the Dead Sea surface temperature. These data showed a statistically significant positive trend of 0.06 °C year$^{-1}$ for daytime SST (increase of 1 °C in SST during the 17 year period under investigation from January 2000 to December 2016) (Fig. 2 a and b; and Table 1). In addition, MODIS data showed a statistically significant positive trend of 0.04 °C year$^{-1}$ for nighttime SST (Fig. 3 a and b;

and Table 1). Note that, in the absence of solar radiation at night, MODIS showed an equal statistically significant positive trend of 0.04 $^{\circ}$C year$^{-1}$ in land skin temperature over the land area in the vicinity of the Dead Sea (Fig. 3 c and d; and Table 1). By contrast to the nighttime period, MODIS data showed the absence of any noticeable trend in daytime land skin temperature (Fig. 2 c and d; and Table 1).

It is noteworthy that this positive daytime SST trend was observed in the absence of positive trend in surface solar radiation, based on pyranometer buoy measurements (Fig. 4). Monthly variations of daily average surface solar radiation (SR) over the 9-year period from 2005 to 2013 revealed even statistically significant negative trend (Fig. 4 a and b, Table 1). Furthermore, the monthly data of daily maximum solar radiation during the same 9-year period revealed no statistically significant trend (Fig. 4 c and d, Table 1).

**3.2 Analysis of factors contributing to the Dead Sea SST trends**

The fact that the positive daytime SST trend was observed in the absence of positive trend in surface solar radiation (based on pyranometer measurements from the hydrometeorological buoy anchored in the Dead Sea) indicates that the observed positive trend in the daytime Dead Sea SST cannot be explained by long-term trends in surface solar radiation.

Neither long-term changes in water mixing in the uppermost layer of the Dead Sea under strong winds, nor long-term
changes in evaporation could explain the observed phenomenon. Indeed, the monthly data of daily average near surface wind speed (based on buoy measurements during the 10-year period from 2005 to 2014) did not reveal any statistically significant trend (Fig. 5 c and d; Table 1). In addition to wind measurements taken at the hydrometeorological buoy anchored in the Dead Sea, we analyzed monthly data of near surface wind speed from two other meteorological stations, located in the vicinity of the Dead Sea: Sdom (31.03N, 35.39E) and Ein-Gedi-SPA (31.42N; 35.38E) (Fig. 1). There were no statistically
significant trends in wind speed taken at the two aforementioned monitoring sites (Fig. 6 and Table 1). Therefore, long-term changes in water mixing in the Dead Sea could not cause the observed positive SST trends.

Furthermore, the observed positive trend in SST could not be explained by a decrease in evaporation. The evaporation process is accompanied by absorption of the latent heat of evaporation, consequently, by a decrease in sea surface temperature. Measured changes in Dead Sea water levels showed a steady decrease during the period from 1993 to 2016
(Fig. 7a). As shown in Fig. 7b, the estimated rate of water level changes from year to year reveals an accelerating decrease in the Dead Sea water level, in accordance with the obtained statistically significant linear fit (Fig. 7b and Table 1). This accelerating decrease in the Dead Sea water level cannot be explained by a steady decrease in water inflow to the Dead Sea from the Jordan River. After the construction of water supply projects in Israel (1964), Jordan (1966) and Syria (1970), the main flow of water into the Jordan River from the Sea of Galilee and from the Yarmouk River was blocked (Holtzman et al.,
2005). Since that time, the only flow of fresh surface water into the Jordan River has included rare flood events and negligible contributions from small springs (Holtzman et al., 2005). According to Gidon Bromberg (Yale Environment, #360, 2008, https://e360.yale.edu/features/will_the_jordan_river_keep_on_flowing ): in 2008 he mentioned that "massive

water withdrawals for irrigation had created lush areas in the Jordan valley but have reduced the river to a trickle in many spots".

In order to analyse long-term changes in evaporation during the study period, we focused on the summer months when the observed decrease in the Dead Sea water level cannot be caused by precipitation. During the period of 2000 – 2001, Holtzman et al. (2005) measured the Jordan River flow rate of approximately 0.5 - 1 $m^3 s^{-1}$. Therefore, even at the beginning of our study period, in every summer month, the water inflow from the Jordan River to the Dead Sea was approximately 1 - 2 x $10^6 m^3$. This is less than 2% of the amount of Dead Sea water loss per month of approximately 100 x $10^6 m^3$ (see Section 4 "SST trends and Dead Sea level drops in the summer months"). Therefore, in the summer months, evaporation is the main contributor to the observed decrease in the Dead Sea water level. Consequently, measured long-term changes in Dead Sea water levels (Fig. 7 a and b) reflect changes in Dead Sea evaporation. This suggests a long-term increase in the Dead Sea evaporation, and this long-term increase is expected to be accompanied by a long-term decrease in sea surface temperature. However, despite the above-mentioned increase in Dead Sea water evaporation, MODIS shows statistically significant positive sea surface temperature trends in both daytime (0.06 $^o$C year$^{-1}$) and nighttime (0.04 $^o$C year$^{-1}$) (Table 1). This indicates the presence of steadily increasing sea surface warming which causes the observed positive SST trends and also compensates surface water cooling due to the increasing evaporation.

The above mentioned steadily increasing sea surface warming suggests some imbalance between incoming and outgoing surface heat flows. To describe the incoming and outgoing surface heat flows, we analyzed temperature differences between Dead Sea SST and land skin temperature, based on MODIS skin temperature measurements. Table 2 represents 17-year monthly means of sea surface temperature (SST) and land skin temperature (LST) in both winter (January) and summer (July). The land skin temperature was averaged over the following two land areas: 1) the land area adjacent to the Dead Sea (between the red boxes and the Dead Sea coastline (Fig. 1)) (LST1), and 2) the land area covered by the red boxes (LST2). One can see that, in the daytime both in summer and in winter: SST < LST1 < LST2, indicating surface horizontal heat transfer from land to sea. The most strong daytime heat transfer from land to sea exists in summer, when the maximum temperature difference of approximately 9 $^o$C is observed between LST1 and SST, compared to that of only 2 $^o$C in winter (Table 2). By contrast to the daytime, in the nighttime: SST > LST1 > LST2, indicating heat transfer from sea to land. The most strong nighttime heat transfer from sea to land exists in winter, when the maximum temperature difference of approximately 5 $^o$C is observed between SST and LST1, compared to that of only 1 $^o$C in summer (Table 2). The aforementioned temperature differences between Dead Sea SST and land skin temperature in the vicinity of the Dead Sea is evidence of the presence of two opposing surface heat flows: from land to sea in the daytime and from sea to land in the nighttime.

Furthermore, there is a positive feedback loop between the positive SST trends and the shrinking of the Dead Sea (Fig. 8). As mentioned in the Introduction, during the period from 1972 to 2013, the Dead Sea water area shrank on average at the rate of ~2.9 $km^2$ year$^{-1}$ (El-Hallaq and Habboub, 2014). This steady reduction of the water area leads to increasing warming of Dead Sea surface water every year by heat flow from land to sea (Fig. 8). Specifically, the surface heat flow from land to

sea (which is proportional to the perimeter of the Dead Sea) heated the steadily shrinking Dead Sea water area. As the reduction of the Dead Sea water area is relatively higher than that of the Dead Sea perimeter (e.g., in the case of a circle: its perimeter is proportional to the radius, while its area is proportional to the square of the radius), this leads to some additional heating of Dead Sea surface water every year and, consequently, to an increase in SST. The increase in SST causes an increase in evaporation. In turn, this contributes to some additional decrease in Dead Sea water levels, eventually to subsequent shrinking of the Dead Sea water area (Fig. 8). This positive feedback loop contributes to the observed statistically significant accelerating rate of the decrease in Dead Sea water levels during the period under consideration (Fig. 7b and Table 1).

Note that positive trends have been detected in air temperature over Israel over several past decades (Yosef et al., 2018). Over a limited area such as the Dead Sea valley this atmospheric warming is uniform. The atmospheric warming uniformly heats the Dead Sea surface water and, consequently, increases SST every year (Fig. 8). In turn, the increased SST leads to an increase in evaporation, contributing to the steady shrinking of the Dead Sea. However, as discussed in the following Section 4, the above-mentioned uniform heating of Dead Sea surface water by the regional atmospheric warming cannot explain the observed non-uniform 17-year mean SST distribution in the summer months.

## 4 SST trends and Dead Sea level drops in the summer months

To support our main finding (positive trends in SST caused by the steadily shrinking Dead Sea) we focused on the summer months for the following reasons: 1) precipitation does not occur; 2) water inflow from the Jordan River is insignificant; and 3) daytime surface heat flows from land to sea are maximal. To this end, we analyzed monthly variations of the Dead Sea water-level drop, estimated separately for each month. This Dead Sea water-level drop was estimated as the difference between the measured Dead Sea water level in the given month and that in the previous month. This was carried out using available monthly measurements of Dead Sea water levels over the 17-year study period of 2000 – 2016, when SST trends were obtained.

In accordance with the obtained 17-year mean seasonal variations of Dead Sea water-level drops, a pronounced maximum was observed in the three consecutive summer months: July, August and September (JAS) (Fig. 9). In particular, in every summer (JAS) month, on average, Dead Sea water level dropped by approximately 0.15 m (Fig. 9). During the study period of 2000 – 2016, the square of the Dead Sea water area is approximately 600 - 640 km$^2$ (El-Hallaq and Habboub, 2014). Consequently, every summer month the Dead Sea loses approximately 100 x 10$^6$ m$^3$ of water. As mentioned in Section 3.2, in these summer months, evaporation significantly contributes to the decrease in the Dead Sea water level (in line with available measurements of Dead Sea evaporation by Metzger et al. (2018, their Fig. 5)), while the contribution of water inflow from the Jordan River is insignificant. Therefore, the obtained summer maximum of the 17-year mean seasonal variations of Dead Sea water-level drop (Fig. 9) is determined mainly by evaporation.

Year-to-year variations of MODIS-based daytime SST, averaged over July, August and September, showed a positive statistically significant trend, during the study period of 2000 - 2016 (Fig. 10a). During the same 17-year period, year-to-year

variations of Dead Sea water-level drop, estimated separately for each summer month and averaged over the JAS summer months, showed a statistically significant trend according to the linear fit: in each of the summer month in 2000 the Dead Sea water level dropped by 0.12 m , while in 2016 by 0.18 m (Fig. 10b). Therefore, in the summer months, in the absence of precipitation and the insignificant water inflow from the Jordan River, some acceleration in the Dead Sea water-level drop was observed during the study period corresponding to the steadily increasing evaporation (Fig. 10b). The increase in evaporation from year to year is caused by the steady warming of Dead Sea surface water (as a result of Dead Sea shrinking) and by a positive feedback loop between the positive SST trends and the shrinking of the Dead Sea. In addition, the observed atmospheric warming heats the Dead Sea surface water causing an increase in evaporation. Thus, during the study period, in the Dead Sea, the steadily increasing heat flow from land to sea together with atmospheric warming is a causal factor of additional evaporation.

A comparison between the above-mentioned temporal variations in daytime SST and those of Dead Sea water-level drop, averaged over the JAS summer months, reveals that these variations are inversely correlated: local maxima of SST coincide with the local minima of the Dead Sea water-level drop (Fig. 10c). This inverse relationship indicates the presence of a positive feedback loop between the shrinking of the Dead Sea and increasing SST trends in the JAS summer months: the higher the SST, the stronger the Dead Sea water-level drop leading to subsequent shrinking of the Dead Sea water area followed by additional surface heating of Dead Sea surface water.

Satellite-based SST measurements showed that the 17-year mean distribution of daytime Dead Sea surface temperature (averaged over the JAS summer months) is non-uniform: maximal SST was observed near the coastline, while minimal SST was observed in the middle of the Dead Sea (Fig. 11). This non-uniform SST distribution indicates that the strongest heating of the surface water takes place on the periphery of the lake and not in the middle. Such a non-uniform SST distribution demonstrates the presence of heat flow from land to sea. The uniform heating of Dead Sea surface water by the regional atmospheric warming cannot explain such non-uniformity in daytime SST.

Shown in Figure 12, a comparison of the spatial distribution of daytime Dead Sea surface temperature (averaged over the JAS summer months) between the two years: 2000 and 2016 illustrates changes in SST during the study period. We found that the average Dead Sea surface temperature increased from 30 $^{\circ}$C in 2000 to 31.3 $^{\circ}$C in 2016. One can see that the most significant increase in SST was observed over the north-west and southern sides of the Dead Sea (Fig. 12), where shrinking of the Dead Sea water area was pronounced (Fig. 13b). This fact of the non-uniform heating of Dead Sea surface water cannot be explained by the uniform atmospheric warming.

Furthermore, such non-uniform heating of Dead Sea surface water was characterized by the non-uniform spatial distribution of long-term SST trends during the study period (Fig. 13a). Maximal SST trends of over 0.8 $^{\circ}$C decade$^{-1}$ were observed over the north-west and southern sides of the Dead Sea, where shrinking of the Dead Sea water area was pronounced (Figs. 13 a and b). No noticeable SST trends were observed over the eastern side of the lake, where shrinking of the Dead Sea water area was insignificant (Figs. 13 a and b). Thus, satellite-based SST measurements showed correspondence between the location of maximal SST trends and that of Dead Sea shrinking: this indicates a causal link between them. This fact implies

the following point: any meteorological, hydrological or geophysical process causing the steady shrinking of the Dead Sea also contributes to the positive trends in SST. This is in accordance with the existing positive feedback loop between the positive SST trends and the shrinking of the Dead Sea, as discussed in Section 3. The positive feedback loop is a causal factor of the observed non-uniform spatial distribution of long-term SST trends during the study period (Fig. 13a).

## 5 Conclusions

In our study, long-term trends in Dead Sea surface temperature (SST) were analyzed using MODIS satellite data of skin surface temperature during the 17-year period of records (2000 – 2016). MODIS data showed positive trends of 0.6 $^{\circ}$C decade$^{-1}$ in the daytime and 0.4 $^{\circ}$C decade$^{-1}$ in the nighttime. These positive SST trends were observed in the absence of

positive trends in surface solar radiation measured by the Dead Sea buoy pyranometer. Therefore the observed increase in the Dead Sea SST over the study period cannot be related to increasing surface solar radiation. We also show that long-term changes in water mixing in the uppermost layer of the Dead Sea under strong winds could not explain the observed SST trends. There is a positive feedback loop between the positive SST trends and the shrinking of the Dead Sea, which contributes to the accelerating decrease in Dead Sea water levels during the period under study

Our study shows that the observed positive SST trends are caused by two factors: increasing daytime heat flow from land to sea (as a result of the steady shrinking of the Dead Sea) and regional atmospheric warming. Positive trends have been detected in air temperature over Israel over several past decades (Yosef et al., 2018). Over the limited area such as the Dead Sea valley this atmospheric warming is uniform. The atmospheric warming uniformly heats the Dead Sea surface water and, consequently, increases SST. In turn, the increased SST leads to an increase in evaporation contributing to the shrinking of

the Dead Sea. However, the uniform heating of Dead Sea surface water by the regional atmospheric warming cannot explain the observed non-uniform 17-year mean SST distribution in the summer months: this non-uniformity is characterized by maximal SST observed near the coastline, while minimal SST was observed in the middle of the Dead Sea. This non-uniform SST distribution demonstrates that the strongest heating of the surface water takes place on the periphery of the lake and not in the middle: this is observational evidence of the presence of heat flow from land to sea. Moreover, satellite-based

SST measurements showed a non-uniform spatial distribution of long-term SST trends during the study period. Maximal SST trends of over 0.8 $^{\circ}$C decade$^{-1}$ were observed over the north-west and southern sides of the Dead Sea, where shrinking of the Dead Sea water area was pronounced (Figs. 13 a and b). No noticeable SST trends were observed over the eastern side of the lake, where shrinking of the Dead Sea water area was insignificant. This finding demonstrates correspondence between the positive SST trends and the shrinking of the Dead Sea: this indicates a causal link between them.

There are two opposite processes taking place in the Dead Sea: sea surface warming and cooling. On the one hand, the positive feedback loop leading to sea surface warming every year accompanied by long-term increase in SST; on the other hand, the measured acceleration of the Dead Sea water-level drop suggests a long-term increase in Dead Sea evaporation accompanied by a long-term decrease in SST. During the period under investigation, the total result of these two opposite

processes is the statistically significant positive SST trends in both daytime and nighttime, observed by MODIS. Therefore, increasing warming of steadily shrinking Dead Sea surface water compensates for surface water cooling (due to increasing evaporation) and even causes observed positive Dead Sea surface temperature trends.

Our findings of the existence of a positive feedback loop between the positive SST trends and the shrinking of the Dead Sea imply the following significant point: any meteorological, hydrological or geophysical process causing the steady shrinking of the Dead Sea will contribute to positive trends in SST.

Our results shed light on continuing danger to the Dead Sea and its possible disappearance. Moreover, it is worth mentioning that shrinking at alarming rates was detected for many of the world's saline lakes which are located in mostly arid areas (Wurtsbaugh et al., 2017). Therefore, our approach can be appropriate for analyzing similar processes in those shrinking saline lakes.

**Data availability and acknowledgements**

Thanks are due to the MODIS teams (PI Name: Zhengming Wan) that produced the LST data. The Collection-6 of the MODIS MOD11C3 Level 3 LST data product DOI: 10.5067/MODIS/MOD11C1.006 was retrieved from the online Data Pool, courtesy of the NASA Land Processes Distributed Active Archive Center (LP DAAC), USGS/Earth Resources Observation and Science (EROS) Center, Sioux Falls, South Dakota, https://lpdaac.usgs.gov/dataset_discovery/modis/modis_products_table/mod11c1_v006#tools

The following data are included in the file with supplementary material uploaded separately: (1) measurements of Dead Sea water levels; (2) monthly data of pyranometer measurements of surface solar radiation from a hydrometeorological buoy, anchored in the Dead Sea; and (3) monthly data of near-surface wind speed measurements from a hydrometeorological buoy, anchored in the Dead Sea. Credit for the buoy data is given to the Israel Oceanographic and Limnological Research. Credit for the data of Dead Sea water levels is given to Israel Hydrological Service. We thank the Israel Meteorological Service for their measurements of near surface wind speed from two meteorological stations located in the vicinity of the Dead Sea: Sdom and Ein-Gedi-SPA , during the study period (http://www.ims.gov.il/IMS/tazpiot/ArchiveTazpiot/).

This study was carried out in the framework of the DESERVE (DEad SEa ResearchVenue) project (https://www.deserve-vi.net/). This project was aimed at studying coupled lithospheric, hydrological, and atmospheric processes in the Dead Sea region (Kottmeier et al., 2015). The work of the Tel Aviv University team was supported by the international Virtual Institute DESERVE funded by the German Helmholtz Association. The work of R.T.P. was supported under grant NNH12ZDA001N-MEASURES from NASA to JPL.

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

**Table 1.** The slope (α) of the obtained linear fit of deseasonalized monthly anomalies of surface temperature over sea (SST) and land (LST) in the daytime (day) and nighttime (night); daily average and daily maximum surface solar radiation ($SR_{DAVE}$ and $SR_{DMAX}$); and daily average near surface wind speed (WS). For daily average near surface wind speed, we used monthly data from the hydrometeorological buoy (WSbuoy) and two meteorological stations: Sdom (WSsdom) and Ein-Gedi-SPA (WSeg). In addition, the slope (α) of the long-term trend in the rate of Dead Sea level changes from year to year (RateDSL, Fig. 6b) is presented. The decision based on the Shapiro-Wilk normality test for residuals (S-W test) and the significance level (p) is also displayed. If the p value was too high as compared with the 0.05 significance level, the obtained linear fit was considered as statistically insignificant.

| Parameter | Period | α | S–W Test | P |
|---|---|---|---|---|
| $SST_{day}$ ($^{o}C$) | 2000 – 2016 | 0.06 ($^{o}C$ year$^{-1}$) | Normal | 0.001 |
| $SST_{night}$ ($^{o}C$) | 2000 – 2016 | 0.04 ($^{o}C$ year$^{-1}$) | Normal | 0.006 |
| $LST_{day}$ ($^{o}C$) | 2000 – 2016 | 0.00 ($^{o}C$ year$^{-1}$) | Normal | Not significant |
| $LST_{night}$ ($^{o}C$) | 2000 – 2016 | 0.04 ($^{o}C$ year$^{-1}$) | Normal | 0.007 |
| $SR_{DAVE}$ (W m$^{-2}$) | 2005 – 2013 | -0.80 (W m$^{-2}$ year$^{-1}$) | Normal | 0.020 |
| $SR_{DMAX}$ (W m$^{-2}$) | 2005 – 2013 | 1.20 (W m$^{-2}$ year$^{-1}$) | Normal | Not significant |
| WSbuoy (m s$^{-1}$) | 2005 – 2014 | 0.00 (m s$^{-1}$ year$^{-1}$) | Normal | Not significant |
| WSsdom (m s$^{-1}$) | 2004 – 2016 | 0.00 (m s$^{-1}$ year$^{-1}$) | Normal | Not significant |
| WSeg (m s$^{-1}$) | 2006 – 2016 | 0.001 (m s$^{-1}$ year$^{-1}$) | Normal | Not significant |
| RateDSL (m year$^{-1}$) | 1992 - 2016 | -0.02 (m year$^{-2}$) | Normal | 0.001 |

**Table 2.** Long-term (17 year) monthly means of sea surface temperature (SST) and land skin temperature (LST) together with their standard deviation (SD) in winter (January) and summer (July). The land skin temperature was averaged over the following two land areas: 1) the land area adjacent to the Dead Sea (between the red boxes and the Dead Sea coastline (Fig. 1)) (LST1), and 2) the land area covered by the red boxes (LST2).

| Month | SST ± SD ($^{o}$C) | LST1 ± SD ($^{o}$C) | LST2 ± SD ($^{o}$C) |
|---|---|---|---|
| | Daytime | | |
| January | 20.4 ± 1.0 | 22.3 ± 1.6 | 22.7 ± 1.9 |
| July | 33.1 ± 0.5 | 41.9 ± 0.6 | 45.9 ± 0.5 |
| | Nighttime | | |
| January | 18.4 ± 1.4 | 13.6 ± 1.4 | 10.9 ± 1.2 |
| July | 31.8 ± 0.5 | 30.5 ± 0.6 | 28.8 ± 0.7 |

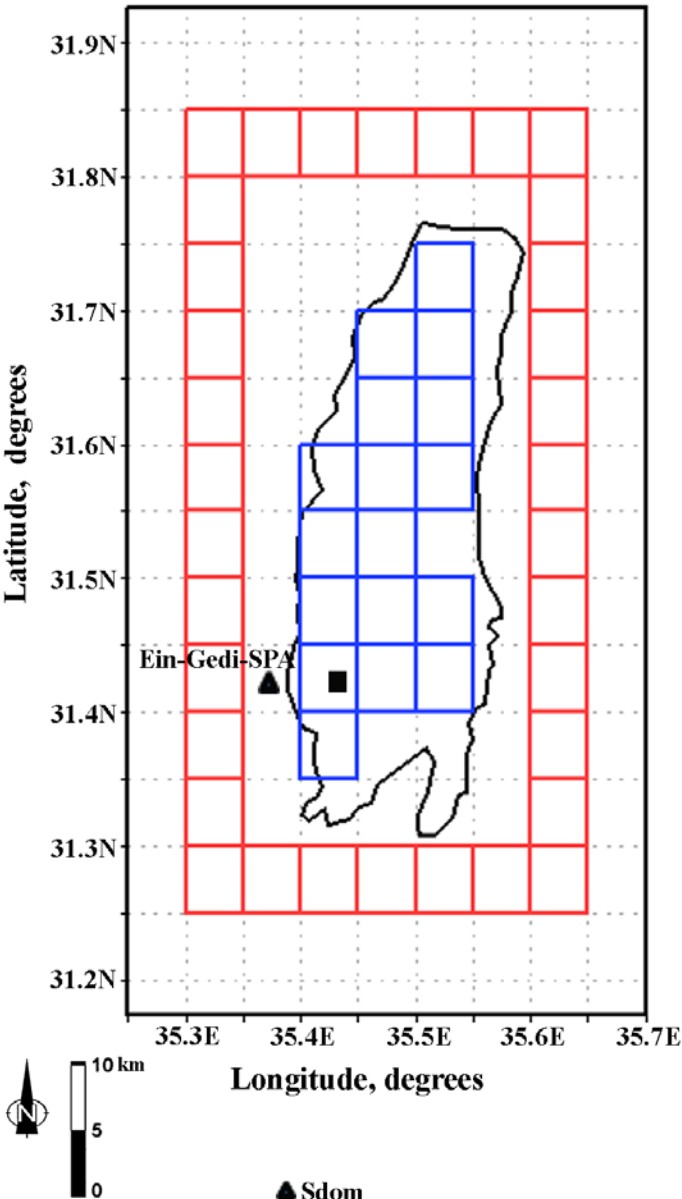

Figure 1: Map of the region under study. The 17 blue boxes show pixels covering the Dead Sea surface. The 34 red boxes show pixels which cover the land area in the vicinity of the Dead Sea. The black square shows the location of the Dead Sea hydrometeorological buoy (31.42$^{o}$N, 35.44$^{o}$E), while the black triangles show the location of two meteorological stations: Sdom (31.03$^{o}$N, 35.39$^{o}$E) and Ein-Gedi-SPA (31.42$^{o}$N; 35.38$^{o}$E).

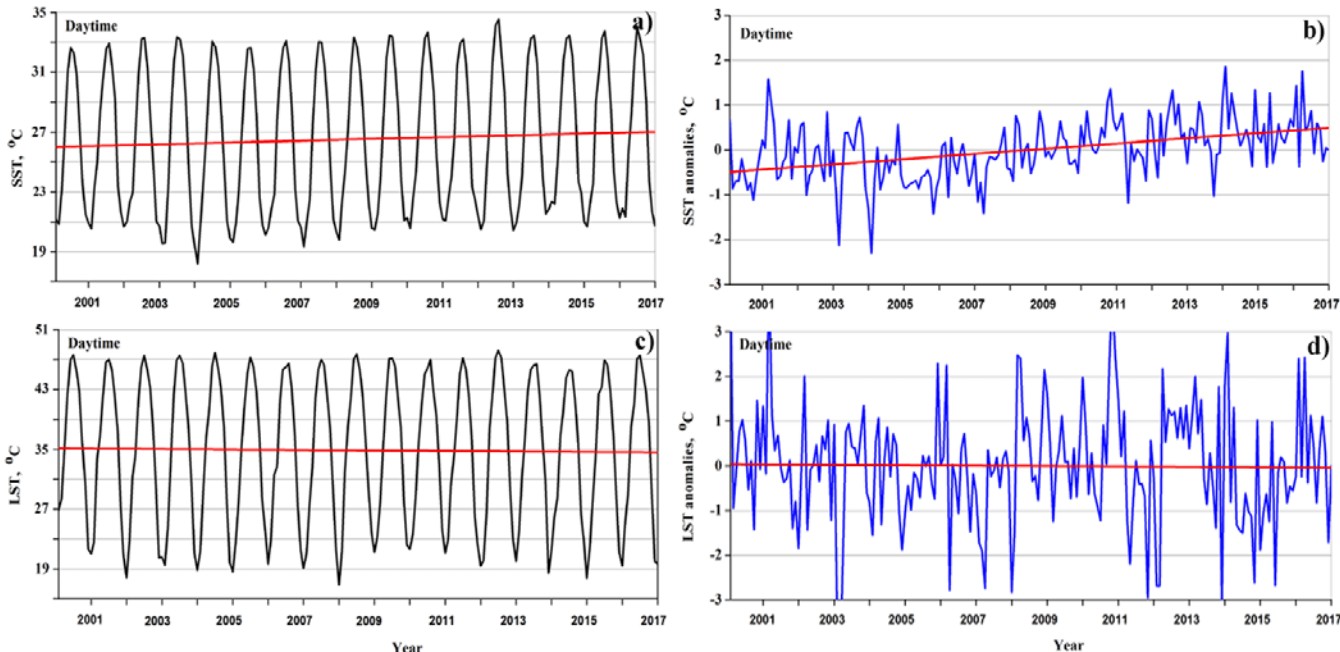

**Figure 2: Daytime monthly variations of (a and b) Dead Sea surface temperature (SST) together with (c and d) land skin temperature (LST) over the land area in the vicinity of the Dead Sea, during the 17-year period under study. The left column represents original MOD11C3 Level 3 monthly data (averaged over the specified sea and land areas), while the right column represents their associated deseasonalized monthly anomalies. The red straight lines designate linear fits.**

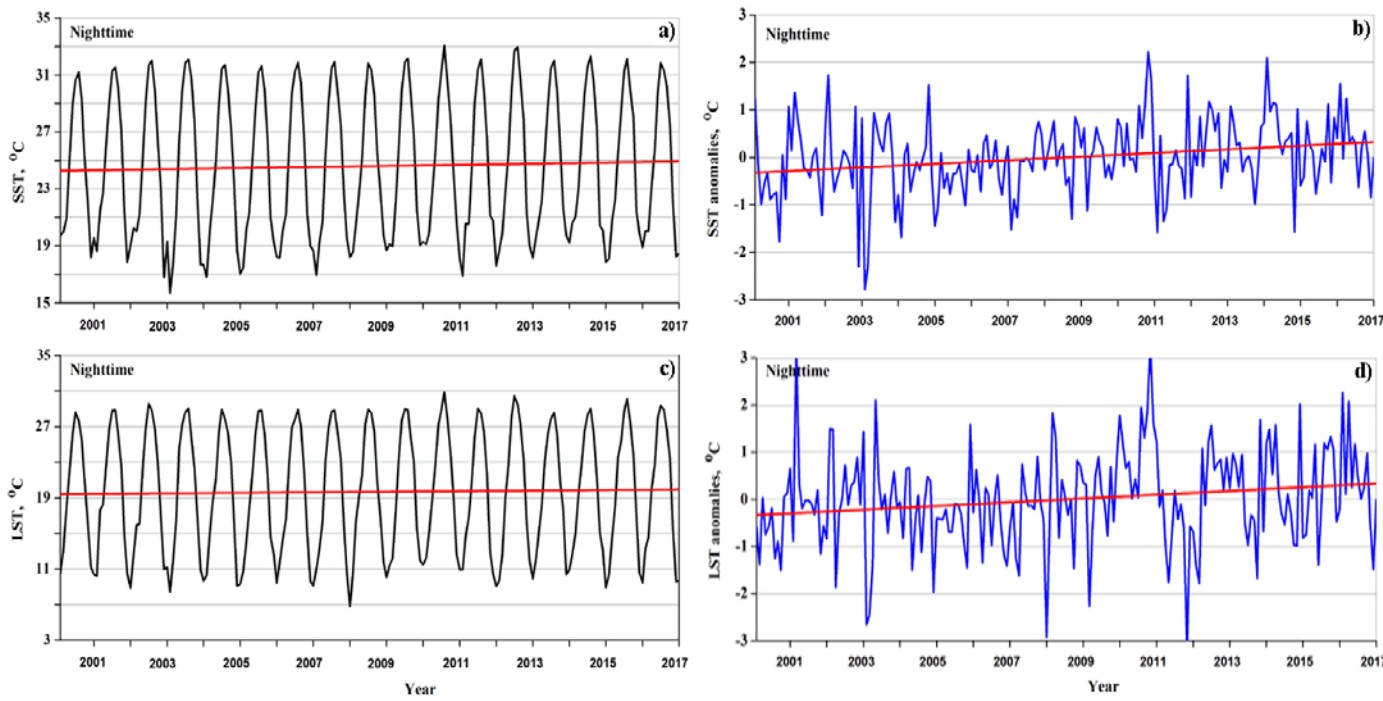

**Figure 3: Nighttime monthly variations of (a and b) Dead Sea surface temperature (SST) together with (c and d) land skin temperature (LST) over the land area in the vicinity of the Dead Sea, during the 17-year period under study. The designations are the same as in Fig. 2.**

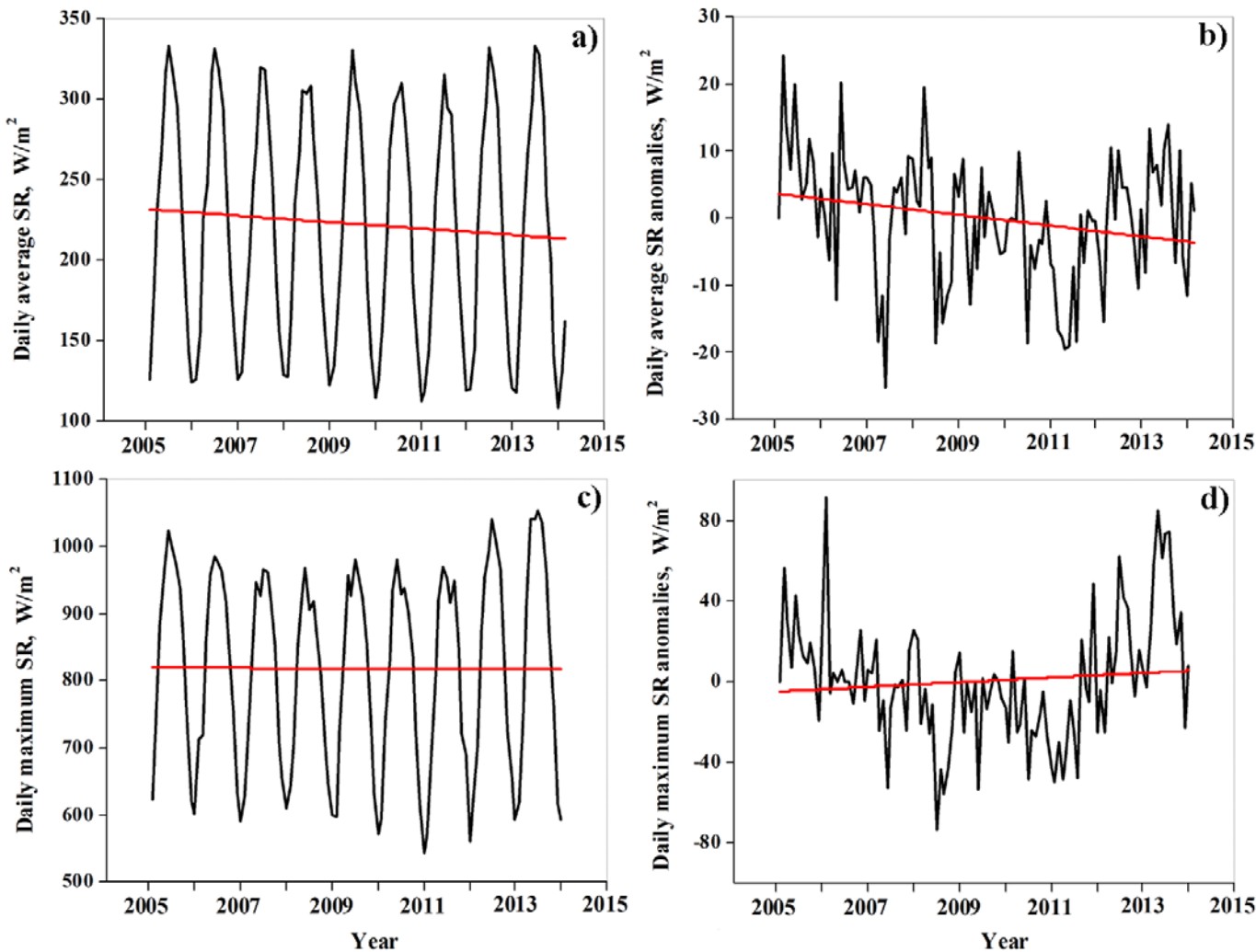

Figure 4: Monthly variations of (a and b) daily average surface solar radiation (SR); and (c and d) daily maximum surface solar radiation, based on pyranometer measurements at the hydrometeorological buoy anchored in the Dead Sea. The left column represents original monthly data, while the right column represents their associated deseasonalized monthly anomalies. The red straight lines designate linear fits.

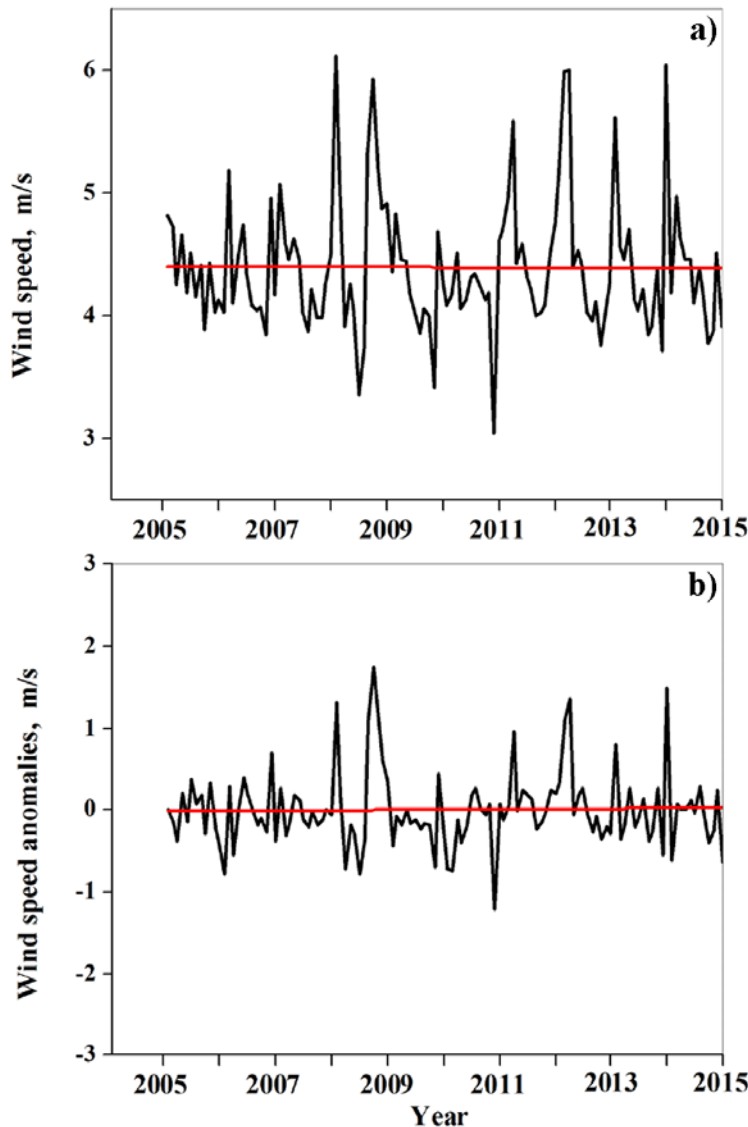

**Figure 5: Monthly variations of daily average near-surface wind speed based on wind measurements at the hydrometeorological buoy anchored in the Dead Sea: a - represents original monthly data, while b - represents their associated deseasonalized monthly anomalies. The red straight lines designate linear fits.**

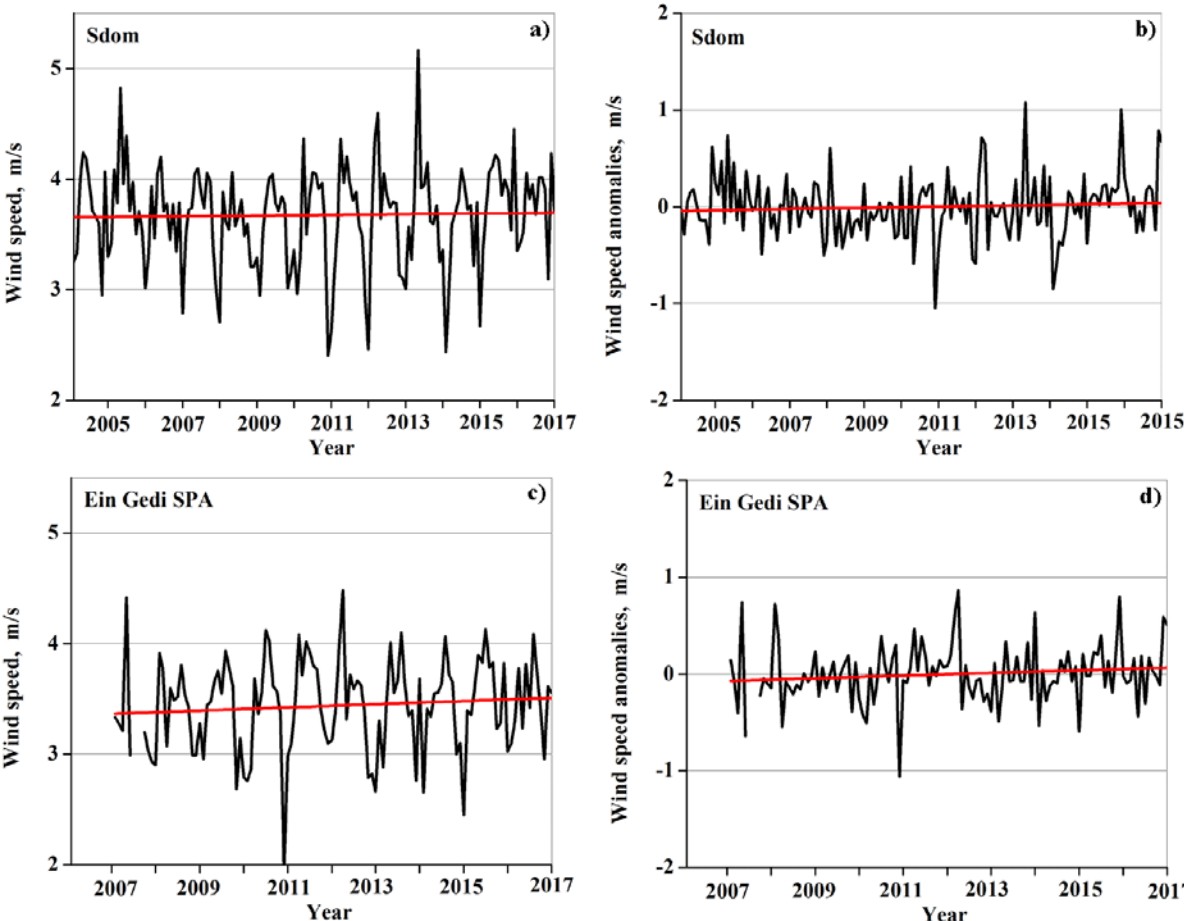

**Figure 6: Monthly variations of daily average near-surface wind speed based on wind measurements at two meteorological stations, located in the vicinity of the Dead Sea: (a and b) Sdom (31.03N, 35.39E) and (c and d) Ein-Gedi-SPA (31.42N; 35.38E). The left column represents original monthly data, while the right column represents their associated deseasonalized monthly anomalies. The red straight lines designate linear fits.**

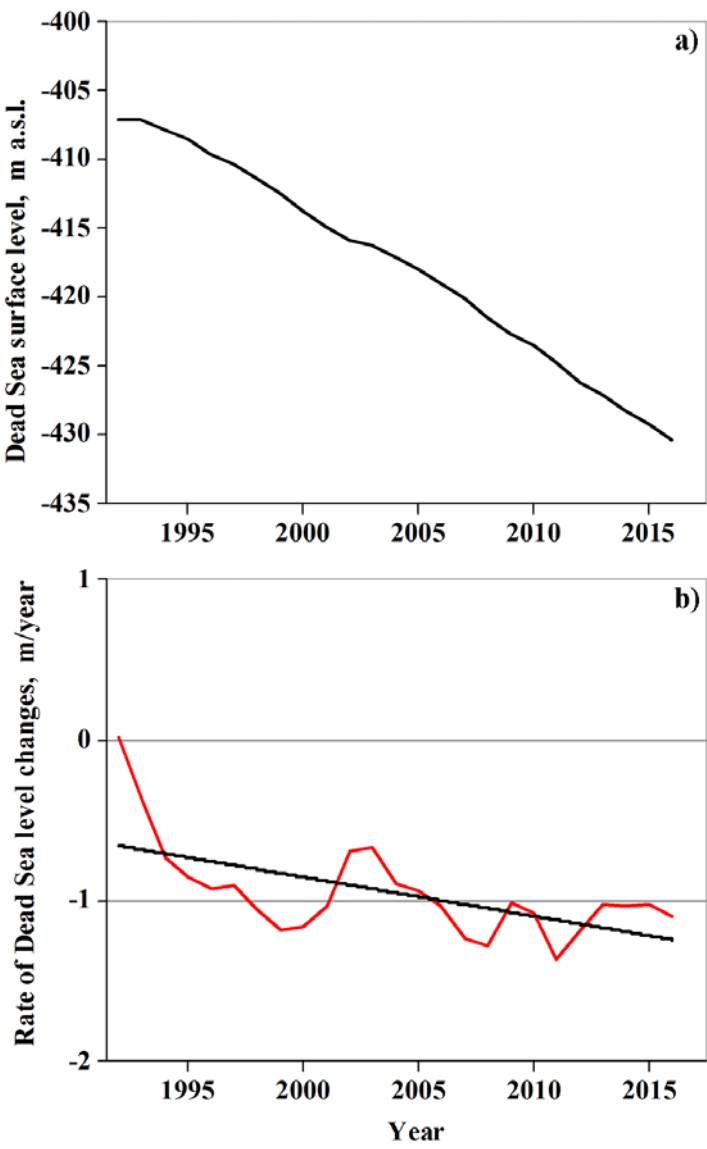

**Figure 7: a - yearly data of the Dead Sea levels (based on available measurements from 1992 until the present); b - the rate of Dead Sea level changes from year to year. The black straight line designates the linear fit.**

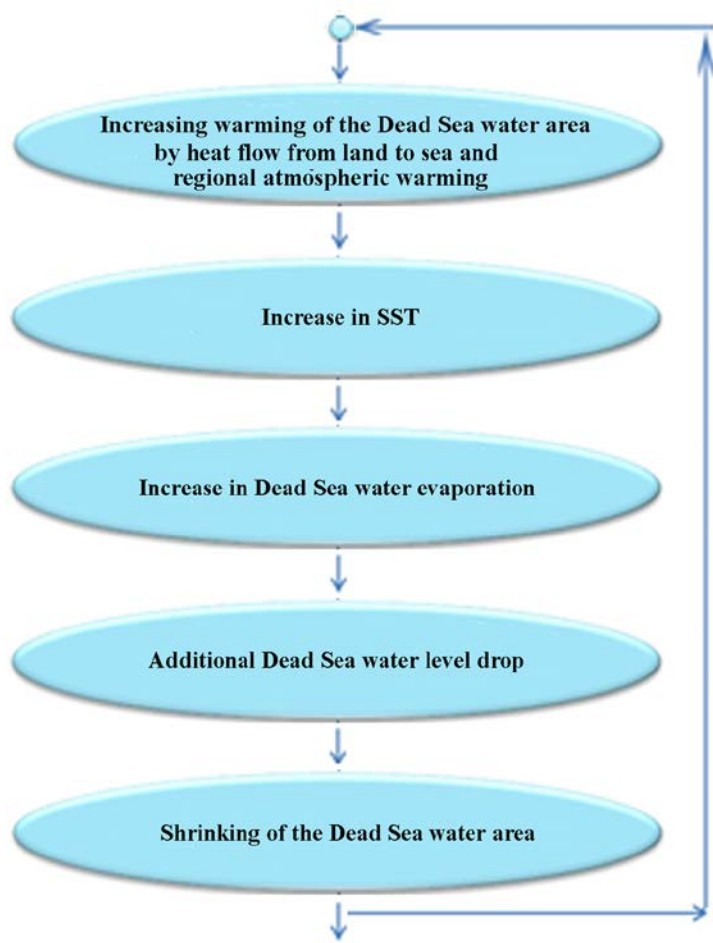

Figure 8. The flowchart of a positive feedback loop between positive Dead Sea surface temperature trends and Dead Sea shrinking.

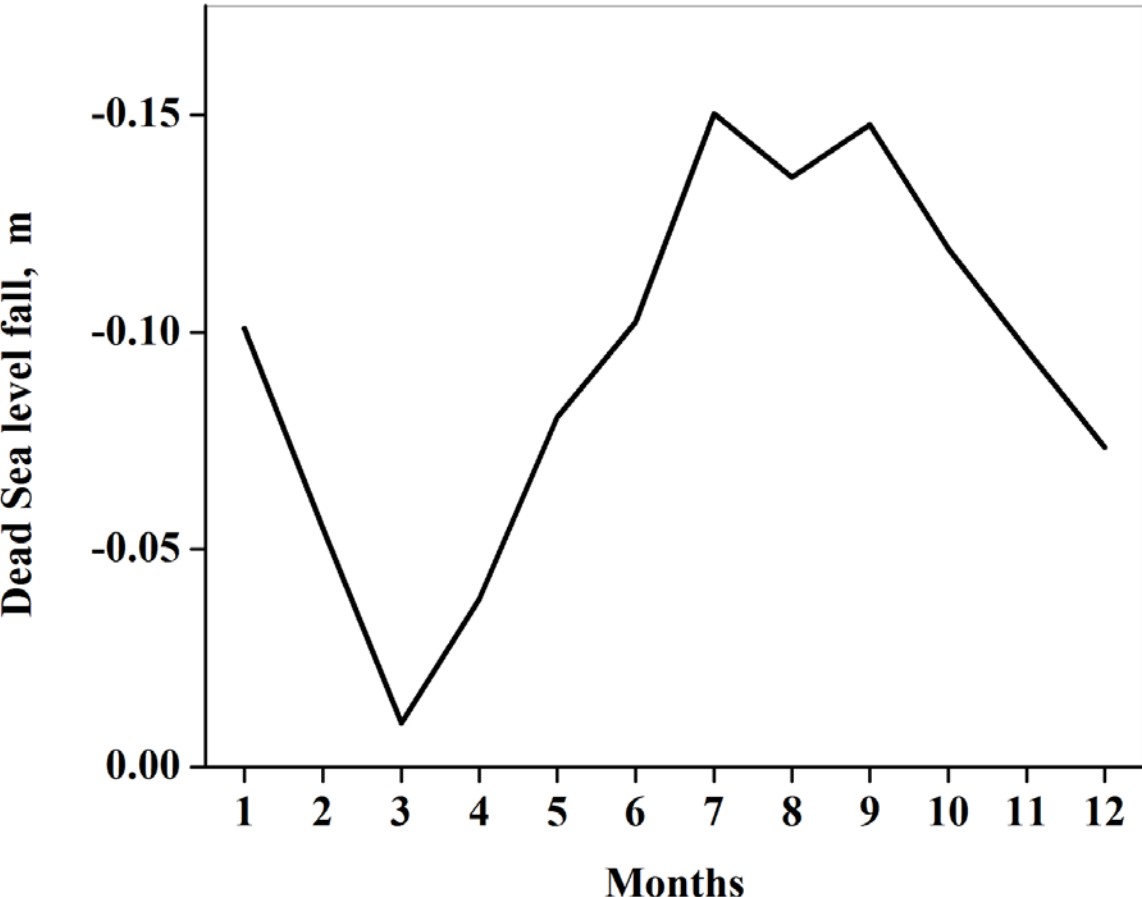

**Figure 9: 17-year (2000 – 2016) mean seasonal variations of the Dead Sea water-level drop estimated separately for each month.**

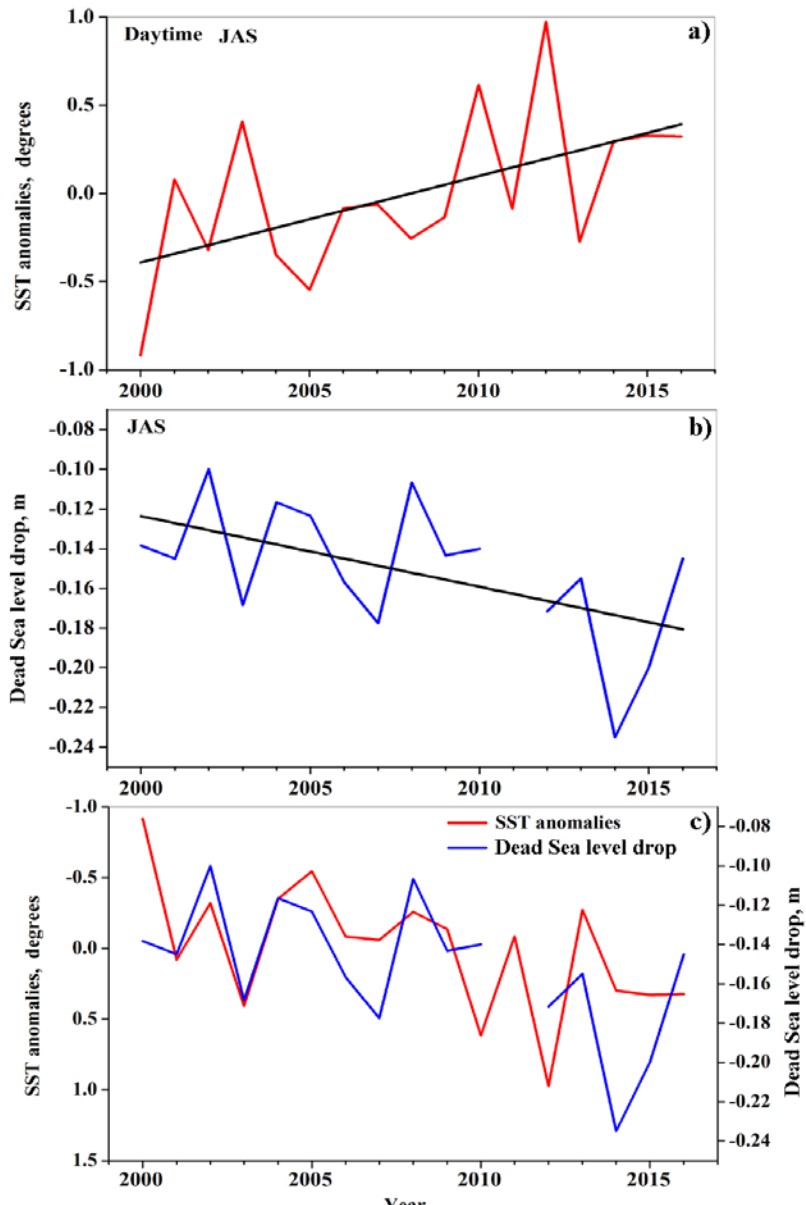

**Figure 10: Year-to-year variations in (a) daytime SST anomalies and (b) Dead Sea water-level drop, averaged over the three summer months July, August, and September (JAS) characterized by the largest Dead Sea water-level drop. c is a comparison between the temporal variations in daytime SST anomalies and Dead Sea water-level drop averaged over the same three summer months. For illustrative purposes, the decreasing scale of SST anomalies is used in (c) instead of the increasing scale used in (a). One can see that these variations are correlated representing an inverse relationship.**

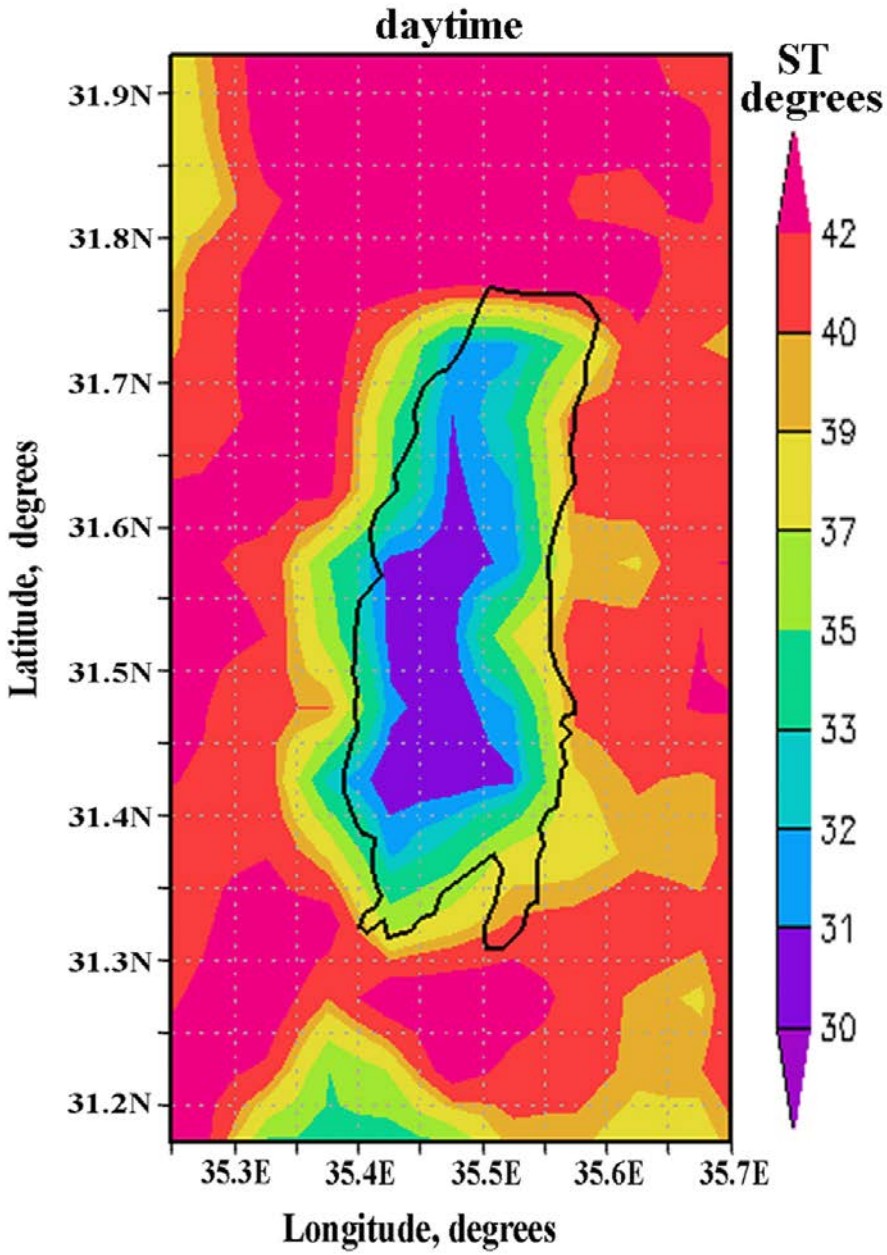

**Figure 11: Spatial distribution of 17-year mean daytime surface temperature (ST, ⁰C) over the Dead Sea and surrounding land areas, averaged over the JAS summer months: July, August, and September.**

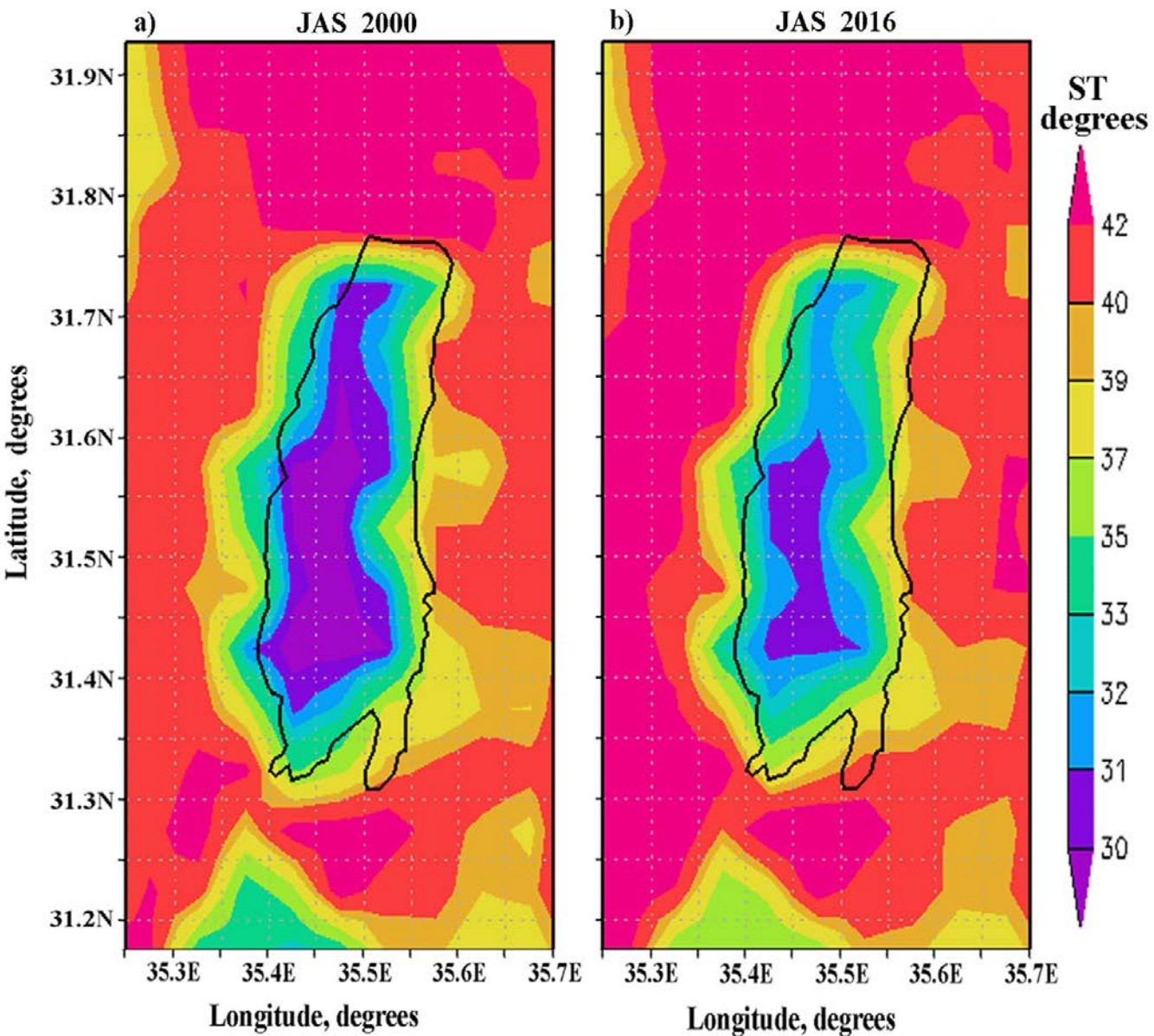

**Figure 12: Comparison of the spatial distribution of daytime surface temperature (ST, ᵒC) over the Dead Sea and surrounding land areas (averaged over the JAS summer months) between the years of 2000 and 2016.**

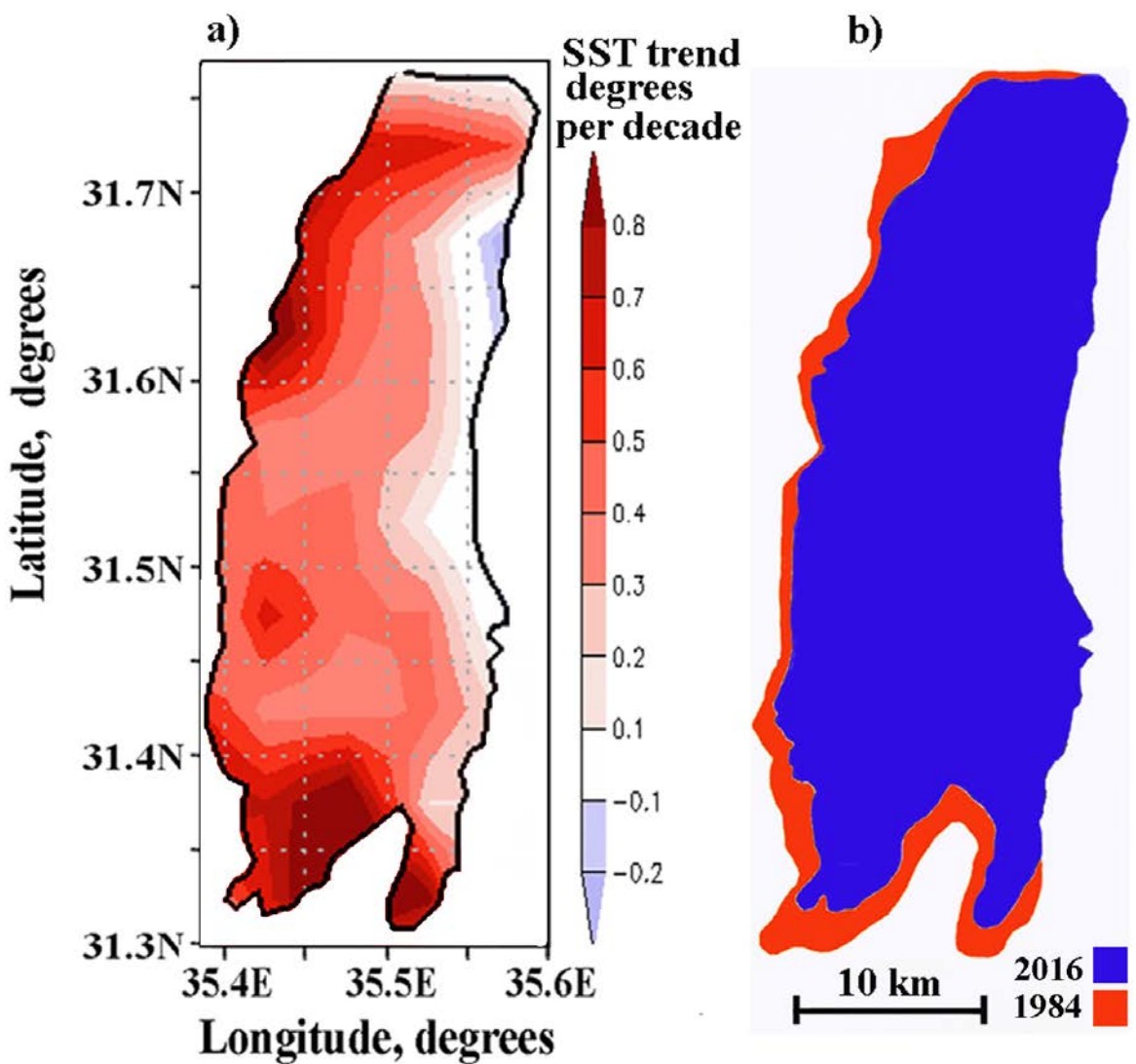

**Figure 13: a - Spatial distribution of 17-year trends of daytime Dead Sea SST (°C decade⁻¹) averaged over the JAS summer months: July, August, and September. b – The Dead Sea water area in the two years: 1984 and 2016, based on data from the Google Earth Engine (Gorelick et al., 2017).**