# Peer review of "Observations of positive sea surface temperature trends in the steadily shrinking Dead Sea"

_Natural Hazards and Earth System Sciences, 2018_

## Referee Comment (RC1) · D. Kaskaoutis (Referee) · 17 Apr 2018

This study analyzes the SST trends over the Dead Sea and surrounding lands using high resolution (3-km) MODIS data. The results show an increasing trend in SST over the Dead Sea, both for daytime and nighttime observations, which is not related with respective positive trends in land skin temperature or even to increase in solar radiation. Authors ascribed this increase in SST temperature to the shrinking of the Dead Sea surface. The manuscript is well written and organized, the results, discussions and explanations are clear enough to any reader. Despite the manuscript is short, it's of good quality due to the knowledge that offers in the scientific community and deserves publication in the journal. I have only some minor comments that have to be corrected. Page 2, Line 23: Revise as "...we used the .." Page 3, Line10: from 2005 to 2013. Page

3, Line 12: Revise as "... was used for the ..." Page 5, Lines 18-19. This sentence is just a repetition of the previous one and can be deleted.

---

## Short Comment (SC1) · 19 Apr 2018

Solid manuscript with no obvious issues or problems.

———————————————————

---

## Referee Comment (RC2) · Anonymous Referee #2 · 22 May 2018

In this paper Kishka et al. use satellite-based Sea Surface Temperature (SST) from a 17-years long record (2000–2016) using MODIS, with the aim of interpreting the long-term changes (trends). The study area is the Dead Sea and surrounding land.

According to the paper, the water level of the Dead Sea dropped significantly in the last 40 years, as a consequence of three factors: 1) decreasing water inflow from the Jordan River; 2) decreasing tendency in rainfall; 3) increasing evaporation.

The authors in this study focuses on the sea surface temperature parameter. Using satellite-based SST, they observe a statistically significant positive trend in Dead Sea at both day time and night time. Previous studies suggest that this increasing SST would be due to increasing surface solar radiation as a result of decreasing cloud cover.

[Figure]

The authors here analyse surface solar radiation measured together with near-surface wind speed from a hydro-meteorological buoy deployed in the Dead Sea. In addition, the author use yearly data of Dead Sea water levels based on available measurements from 1992 until now.

From those pointwise measurements, they find the absence of positive trends in surface solar radiation. This means that the observed positive trend in the daytime SST cannot be explained in terms of surface solar radiation trends. They also do not observe statistically significant trend in near surface wind. They interpret in stable mixing over time (as more mixing would produce more evaporation). Measurements of water level clearly indicate a negative trend over the observational period.

In my opinion, the reasoning of the authors to draw conclusions is uncomplete, i.e., that the evidence that "the observed increase in the Dead Sea SST over the study period cannot be related to increasing surface solar radiation", supports that this SST trend is the net result of two opposite processes: 1) increased evaporation that results in decreased SST in the long-term; 2) reducing water surface that leads to some additional surface heating (increased SST) every year.

However, a number of factors need to be considered:

1) warming of waters produces expansion and the water level should increase. However, this effect is much more evident in big oceans than in shallow inland water bodies that are subjected to marked year to year changes. A warming atmosphere produces more evaporation, meaning more water is available for precipitation.

2) If the level drops, the density and saltiness might rising until a certain point where the rate of evaporation will reach a kind of equilibrium.

3) the evaporation of water depends on open water surface. For instance the water surface area influences the quantity lost through evaporation from water (if the water surface decreases less surface is exposed to atmosphere). The authors look at one

single point that is not enough to draw conclusions about wind effects. This point needs to be better supported (e.g. measuring the water surface area chance over time)

4) Evaporation is influenced by the temperature of air above (higher air temperatures favourite the rate of evaporation). But also the vapor pressure of the liquid has to be considered. The rate of evaporation therefore depends on the difference between saturation vapor pressure at the water temperature and at the dew point of the air. Higher the difference, more the evaporation.

5) Another comment is about the wind. More than amplitude, evaporation is influenced by surface roughness. Moreover, nothing is said about air pressure. I would expect decreasing evaporation with increasing pressure.

6) Another point is that most of the water volume of the Jordan River is extracted before the river reaches the Dead Sea. How much is contributing to the lake drop? It is important to know before saying evaporation is main contribution to the decrease of the water level.

To conclude, the topic is certainly interesting, but authors are missing parts that could influence their conclusions. In my opinion, they have to better describe the occurring meteorological, hydrological, and geophysical processes and their interactions with the related variables that could play a key role. Also, the bibliography is just mentioned and somewhat missing (see below where there is important discussion about solar radiation and wind effects) and should be better explained how some conclusions are stated about water level drops, evaporation, winds, etc.

e.g. Lensky, N. G., Lensky, I. M., Peretz, A., Gertman, I., Tanny, J., & Assouline, S. (2018). Diurnal course of evaporation from the Dead Sea in summer: A distinct double peak induced by solar radiation and night sea breeze. Water Resources Research, 54(1), 150-160.

Overall, the paper calls for some major revision. I like reading the revised version.

---

## Author Comment (AC2) · 12 Jun 2018

Dear Dr. Kaplan,

We thank you for your positive comment on our manuscript.

---

## Author Comment (AC3) · 12 Jun 2018

Dear Reviewer #2,

Below we present our answers (**A**) to your comments (**C**):

> **C**: *"In this paper Kishcha et al. use satellite-based Sea Surface Temperature (SST) from a 17-years long record (2000–2016) using MODIS, with the aim of interpreting the longterm changes (trends). The study area is the Dead Sea and surrounding land.*
>
> *According to the paper, the water level of the Dead Sea dropped significantly in the last 40 years, as a consequence of three factors: 1) decreasing water inflow from the Jordan River; 2) decreasing tendency in rainfall; 3) increasing evaporation.*
>
> *The authors in this study focus on the sea surface temperature parameter. Using satellite-based SST, they observe a statistically significant positive trend in Dead Sea at both day time and night time. Previous studies suggest that this increasing SST would be due to increasing surface solar radiation as a result of decreasing cloud cover.*
>
> *The authors here analyze surface solar radiation measured together with near-surface wind speed from a hydro-meteorological buoy deployed in the Dead Sea. In addition, the authors use yearly data of Dead Sea water levels based on available measurements from 1992 until now .*
>
> *From those pointwise measurements, they find the absence of positive trends in surface solar radiation. This means that the observed positive trend in the daytime SST cannot be explained in terms of surface solar radiation trends. They also do not observe statistically significant trend in near surface wind. They interpret in stable mixing over time (as more mixing would produce more evaporation). Measurements of water level clearly indicate a negative trend over the observational period .*
>
> *In my opinion, the reasoning of the authors to draw conclusions is incomplete, i.e., that the evidence that "the observed increase in the Dead Sea SST over the study period cannot be related to increasing surface solar radiation", supports that this SST trend is the net result of two opposite processes: 1) increased evaporation that results in decreased SST in the long-term; 2) reducing water surface that leads to some additional surface heating (increased SST) every year."*

**A**: Measurements showed two contradictory trends: on the one hand, positive SST trends in the absence of increasing trends in solar radiation and, on the other hand, an accelerating rate of the Dead Sea water level drops (due to an increase in evaporation which is accompanied by a decrease in SST). In our study, we explained these contradictory trends by the presence of the daytime surface heat flow from land to sea. This surface heat flow is maximal in the summer months, when the maximum

temperature difference of approximately 9$^{o}$C between land and sea is observed (Table 2). Moreover, in the summer months, there is no precipitation and the water inflow from the Jordan River to the Dead Sea is minimal. Specifically, in every summer month the water inflow from the Jordan River to the Dead Sea (~1 - 2 x 10$^6$ m$^3$) is insignificant compared to the water loss from the Dead Sea per month (~100 x 10$^6$ m$^3$) (see our reply to the Reviewer's specific comment 6). Therefore, in the summer months, in the absence of precipitation and in the presence of insignificant water inflow from the Jordan River to the Dead Sea, evaporation is the main cause of the Dead Sea water level drop. Consequently, in the summer months, the observed accelerating rate of Dead Sea water level drop corresponds to the long-term increase in evaporation.

As mentioned by Reviewer #2 in his specific comments (below), several factors could influence Dead Sea evaporation. The total result of all these factors is the above-mentioned long-term increase in evaporation, which is expected to be accompanied by the long-term decrease in SST. However, satellite measurements show positive trends in Dead Sea surface temperature. These SST trends are explained in our study as follows.

The daytime surface heat flow from land to sea (which is proportional to the perimeter of the Dead Sea) heats the steadily shrinking Dead Sea water area. As the reduction of the Dead Sea water area is relatively higher than that of the Dead Sea perimeter, this leads to some additional surface heating of Dead Sea water every year. This additional heating of Dead Sea surface water (as a result of Dead Sea shrinking) is leading to an increase in water evaporation, consequently, to some additional decrease in Dead Sea water levels, eventually to subsequent shrinking of the Dead Sea water area. Therefore, there is a positive feedback loop between the shrinking of the Dead Sea and the positive SST trends. The observed shrinking of the Dead Sea water area followed by additional sea surface warming every year is the main process contributing to the observed positive trends in both daytime and nighttime SST. The steadily shrinking Dead Sea, followed by sea surface warming, not only compensates surface water cooling (due to the increasing evaporation) but even causes the observed positive SST trends.

**New Supporting Material**

We are going to add support to our main finding (positive trends in SST caused by the steadily shrinking Dead Sea) by focusing on the summer months, when the contribution of water inflow from the Jordan River is insignificant and the daytime surface heat flow from land to sea is maximal. To this end, we analyzed monthly variations of the Dead Sea water level drop, estimated separately for each month. This Dead Sea water level drop was estimated as the difference between the measured Dead Sea water level in the given month and that in the previous month. This was carried out using available monthly measurements of Dead Sea water levels over the 17-year study period of 2000 – 2016, when SST trends were obtained. This approach allowed us to minimize the influence of water level drops in previous months.

In accordance with the obtained 17-year mean annual variations of Dead Sea water level drops, a pronounced maximum was observed in the three consecutive summer months: July, August and September (a new Figure A1).

[Figure]

*Figure A1. 17-year (2000 – 2016) mean annual variations of the Dead Sea level drop estimated separately for each month.*

[Figure]

Figure A2: *Year-to-year variations in (a) daytime SST anomalies and (b) Dead Sea level drop, averaged over the three summer months July, August, and September (JAS), characterized by the largest Dead Sea water level drop. The black straight lines designate linear fits.*

In these summer months, evaporation significantly contributes to the decrease in the Dead Sea water level (in line with available measurements of Dead Sea evaporation (Metzger et al., 2018, their Fig. 5)), while the contribution of water inflow from the Jordan River is insignificant. Therefore, the obtained summer maximum of the 17-year mean seasonal variations of Dead Sea water level drop (the new Figure A1) is determined mainly by evaporation (see also our reply to the Reviewer's specific comment 6).

Year-to-year variations of MODIS-based daytime SST, averaged over July, August and September, showed a positive statistically significant trend during the study period of 2000 - 2016 (a new Figure A2a). During the same 17-year period, year-to-year variations of Dead Sea water level drop, estimated separately for each summer month and averaged over the JAS summer months, showed a negative statistically significant trend according to the linear fit (a new Figure A2b): in 2000, the water level in one summer month dropped by 0.12 m, while in 2016 by 0.18 m. This negative trend indicates acceleration in evaporation in the summer months. Therefore, in the summer months, the steady shrinking of the Dead Sea, followed by sea surface warming, compensates surface water cooling due to the increasing evaporation, and even causes the observed positive Dead Sea surface temperature trend (Figure A2a).

**Specific comments by Reviewer #2**

1) **C:** *"However, a number of factors need to be considered:*

   *1) warming of waters produces expansion and the water level should increase. However, this effect is much more evident in big oceans than in shallow inland water bodies that are subjected to marked year to year changes. A warming atmosphere produces more evaporation, meaning more water is available for precipitation."*

   **A**: According to the Reviewer's remark, "warming of waters produces expansion and the water level should increase". This process might exist, however it could not prevent the observed acceleration in the long-term Dead Sea water level drop (the new Figure A2b). As for the effect of atmospheric warming producing more evaporation, this is only one of the factors influencing Dead Sea evaporation. As mentioned in our reply to the previous general comment by the Reviewer, the total result of all factors influencing Dead Sea evaporation is the long-term increase in evaporation observed during the study period.

2) **C:** *"If the level drops, the density and saltiness might rising until a certain point where the rate of evaporation will reach a kind of equilibrium."*

   **A**: In the summer months, in the absence of precipitation and the insignificant water inflow from the Jordan River, an acceleration in the long-term Dead Sea water level drop was observed corresponding to the steadily increasing evaporation (the new Figure

A2b). Thus, during the study period, there was no equilibrium in the rate of evaporation. The increase in evaporation from year to year is caused by the steady warming of Dead Sea surface water (as a result of Dead Sea shrinking) and by a positive feedback loop between the shrinking of the Dead Sea and positive SST trends. Thus, during the study period, in the Dead Sea, the steadily increasing heat flow from land to sea is a causal factor of additional evaporation which prevents equilibrium.

3) **C:** *"The evaporation of water depends on open water surface. For instance the water surface area influences the quantity lost through evaporation from water (if the water surface decreases less surface is exposed to atmosphere). The authors look at one single point that is not enough to draw conclusions about wind effects. This point needs to be better supported (e.g. measuring the water surface area chance over time)."*

**A**: The positive SST trends, based on satellite measurements, were observed over the whole area of the Dead Sea (Fig. 1) and not only over one single point. With respect to the Reviewer's remark about the use of wind measurements from only one measuring site, monthly data of wind speed from two other meteorological stations (located in the vicinity of the Dead Sea) were analyzed. We refer the Reviewer to our answer to his specific comment 5.

4) **C:** *"Evaporation is influenced by the temperature of air above (higher air temperatures favorite the rate of evaporation). But also the vapor pressure of the liquid has to be considered. The rate of evaporation therefore depends on the difference between saturation vapor pressure at the water temperature and at the dew point of the air. Higher the difference, more the evaporation."*

**A**: All factors mentioned by the Reviewer (such as air temperature, vapor pressure, the difference between saturation vapor pressure at the water temperature and at the dew point of the air) influence evaporation from the Dead Sea.
In every summer month, the water inflow from the Jordan River to the Dead Sea (~1 - 2 x $10^6$ m$^3$) is insignificant compared to the water loss from the Dead Sea (~100 x $10^8$ m$^3$, see our reply to the Reviewer's specific comment 6). Therefore, in the summer months, evaporation has to be the main cause of water loss from the Dead Sea. The observed accelerating rate of Dead Sea water level drop (the new Figure A2b) is an indication that the total result of all factors influencing Dead Sea evaporation is the long-term increase in evaporation, which is expected to be accompanied by a long-term decrease in SST.

5) **C:** *"Another comment is about the wind. More than amplitude, evaporation is influenced by surface roughness. Moreover, nothing is said about air pressure. I would expect decreasing evaporation with increasing pressure".*

**A**: In accordance with the Reviewer's remark, in addition to wind measurements taken at the hydrometeorological buoy anchored in the Dead Sea, we analyzed monthly data of near surface wind speed from two other meteorological stations, located in the vicinity of the Dead Sea: Sdom (31.03N, 35.39E) and Ein-Gedi-SPA (31.42N; 35.38E). There were no statistically significant trends in wind speed taken at the two aforementioned monitoring sites.

As for the Reviewer's remark about air pressure, Hect and Gertman (2003) discussed a statistically significant positive trend of 1.1 hPa/decade in air pressure, based on their measurements at the hydrometeorological buoy anchored at the Dead Sea, during the 10-year period from 1992 – 2002. They explained this positive trend in air pressure by the fact that the air pressure gauge mounted on the buoy dropped by approximately 7 m, as a result of the Dead Sea water level drop during that period, (Hect and Gertman, 2003). According to the Reviewer's remark, he "would expect decreasing evaporation with increasing pressure". This process might exist, however it did not prevent the acceleration in the long-term Dead Sea water level drop observed during that period of 1992 – 2002 (Fig. 6b).
.

6) **C:** *"Another point is that most of the water volume of the Jordan River is extracted before the river reaches the Dead Sea. How much is contributing to the lake drop? It is important to know before saying evaporation is main contribution to the decrease of the water level."*

**A**: We consider that, in the summer months, evaporation is the main contributor to the observed decrease in the Dead Sea water level. In every summer month (July, August, September), on average, Dead Sea water level dropped by approximately 0.15 m (the new Figure A1). According to El-Hallaq and Habboub (2014), during the study period of 2000 – 2016, the square of the Dead Sea water area is approximately 600 - 640 $km^2$. Consequently, every summer month the Dead Sea loses approximately 100 x $10^6$ $m^3$ of water.

As for the Jordan River, after the construction of water supply projects in Israel (1964), Jordan (1966) and Syria (1970), the main flow of water into the Jordan River from the Sea of Galilee and from the Yarmouk River was blocked (Holtzman et al., 2005). Since that time, the only flow of fresh surface water into the Jordan River has included rare flood events and negligible contributions from small springs. According to Gidon Bromberg (Yale Environment, #360, 2008, https://e360.yale.edu/features/will_the_jordan_river_keep_on_flowing ): in 2008 he mentioned that "massive water withdrawals for irrigation had created lush areas in the Jordan valley but have reduced the river to a trickle in many spots" (see a new Figure A3). During the period of 2000 – 2001, Holtzman et al. (2005) measured the Jordan River flow rate of approximately 0.5 - 1 $m^3$ $s^{-1}$. Therefore, in every summer month, the water inflow from the Jordan River to the Dead Sea was approximately 1 - 2 x $10^6$ $m^3$. This is less than 1% of the above mentioned amount of Dead Sea water loss per month.

[Figure]

*Figure A3. Massive water withdrawals for irrigation have created lush areas in the Jordan valley but have reduced the river to a trickle in many spots. Photo and the caption: Gidon Bromberg (Yale Environment, #360, 2008, (https://e360.yale.edu/features/will_the_jordan_river_keep_on_flowing )*

**C:** *"To conclude, the topic is certainly interesting, but authors are missing parts that could influence their conclusions. In my opinion, they have to better describe the occurring meteorological, hydrological, and geophysical processes and their interactions with the related variables that could play a key role. Also, the bibliography is just mentioned and somewhat missing (see below where there is important discussion about solar radiation and wind effects) and should be better explained how some conclusions are stated about water level drops, evaporation, winds, etc.*

*e.g. Lensky, N. G., Lensky, I. M., Peretz, A., Gertman, I., Tanny, J., & Assouline, S. (2018). Diurnal course of evaporation from the Dead Sea in summer: A distinct double peak induced by solar radiation and night sea breeze. Water Resources Research, 54(1), 150-160.*

*Overall, the paper calls for some major revision. I like reading the revised version."*

**A**: Various meteorological, hydrological, and geophysical processes could influence Dead Sea evaporation (Lensky et al., 2005, 2018; Metzger et al., 2018). To explain the contradictory trends based on available measurements (i.e. the positive SST trends and the accelerating rate of the Dead Sea water level drop due to a long-term increase in evaporation), our study focuses on the total result of all these factors. The total result of all these factors is a long-term increase in evaporation, particularly in the summer months during the study period.

The reference to Lensky et al. (2018) will be added to the revised version, in accordance with the Reviewer's remark.

We thank Reviewer #2 for his comments on our manuscript, which helps us to clarify the paper.

References

Hecht, A., Gertman, I. (2003). Dead Sea meteorological climate, in Nevo, E., Oren, A., Wasser, S.P. (Eds.) Fungal Life in the Dead Sea. Oberreifenberg, Germany, ISBN 978-3-906166-10-0, pp. 68 - 114.

Holtzman, R., Shavit, U., Segal-Rozenhaimer, M., Gavrieli, I., Marei, A., Farber, E., Vengosh, A. (2005). Quantifying ground water inputs along the Lower Jordan River. Journal of Environmental Quality, 34, 897 – 906, doi:10.2134/jeq2004.0244.

Lensky, N.G., Dvorkin, Y., Lyakhovsky, V., Gertman, I., Gavrieli, I. (2005). Water, salt, and energy balances of the Dead Sea, Water Resources Research, 41, W12418, doi:10.1029/2005WR004084.

Metzger, J., Nied, M., Corsmeier, U., Kleffmann, J., Kottmeier, C. (2018). Dead Sea evaporation by eddy covariance measurements vs. aerodynamic, energy budget, Priestley–Taylor, and Penman estimates, Hydrol. Earth Syst. Sci., 22, 1135-1155, https://doi.org/10.5194/hess-22-1135-2018, 2018.

---

## Author Response (AR1)

Dear Dr. Kaskaoutis, Reviewer #1,

Below we present our answers (**A**) to your comments (**C**).

**C**: *This study analyzes the SST trends over the Dead Sea and surrounding lands using high resolution (3-km) MODIS data. The results show an increasing trend in SST over the Dead Sea, both for daytime and nighttime observations, which is not related with respective positive trends in land skin temperature or even to increase in solar radiation. Authors ascribed this increase in SST temperature to the shrinking of the Dead Sea surface. The manuscript is well written and organized, the results, discussions and explanations are clear enough to any*

10 *reader. Despite the manuscript is short, it's of good quality due to the knowledge that offers in the scientific community and deserves publication in the journal. I have only some minor comments that have to be corrected. Page 2, Line 23: Revise as "...we used the .." Page 3, Line10: from 2005 to 2013. Page C1 NHESSD Interactive comment Printer-friendly version Discussion paper 3, Line 12: Revise as "... was used for the ..." Page 5, Lines 18-19. This sentence is just a repetition of the previous one and can be deleted.*

**A**: We thank Dr. Kaskaoutis for his positive review and helpful comments on our manuscript. In the revised version, the text has been updated according to the Reviewer's comments.

Dear Reviewer #2,

Below we present our answers (**A**) to your comments (**C**):

5           **C**: *In this paper Kishcha et al. use satellite-based Sea Surface Temperature (SST) from a 17-years long record (2000–2016) using MODIS, with the aim of interpreting the longterm changes (trends). The study area is the Dead Sea and surrounding land.*

*According to the paper, the water level of the Dead Sea dropped significantly in the last 40 years, as a consequence of three factors: 1) decreasing water inflow from the Jordan River; 2) decreasing tendency in*

10           *rainfall; 3) increasing evaporation.*

*The authors in this study focus on the sea surface temperature parameter. Using satellite-based SST, they observe a statistically significant positive trend in Dead Sea at both day time and night time. Previous studies suggest that this increasing SST would be due to increasing surface solar radiation as a result of decreasing cloud cover.*

15           *The authors here analyze surface solar radiation measured together with near-surface wind speed from a hydro-meteorological buoy deployed in the Dead Sea. In addition, the authors use yearly data of Dead Sea water levels based on available measurements from 1992 until now.*

*From those pointwise measurements, they find the absence of positive trends in surface solar radiation. This means that the observed positive trend in the daytime SST cannot be explained in terms of surface solar*

20           *radiation trends. They also do not observe statistically significant trend in near surface wind. They interpret in stable mixing over time (as more mixing would produce more evaporation). Measurements of water level clearly indicate a negative trend over the observational period.*

*In my opinion, the reasoning of the authors to draw conclusions is incomplete, i.e., that the evidence that "the observed increase in the Dead Sea SST over the study period cannot be related to increasing surface solar*

25           *radiation", supports that this SST trend is the net result of two opposite processes: 1) increased evaporation that results in decreased SST in the long-term; 2) reducing water surface that leads to some additional surface heating (increased SST) every year.*

          **A**: In accordance with the Reviewer's comments, in the revised version, we have included some additional material to

30           support our findings. Measurements showed two opposite trends: on the one hand, positive SST trends in the absence of increasing trends in solar radiation and, on the other hand, an accelerating rate of the Dead Sea water-level drop (due to an increase in evaporation which is accompanied by a decrease in SST). In our study, we explained these opposite trends by the presence of the daytime surface heat flow from land to sea and the steady shrinking of the Dead Sea.

35           Note that the regional atmospheric warming has been measured over Israel over several past decades (Yosef et al., 2018). Over a limited area, such as the Dead Sea valley, this atmospheric warming is uniform. The atmospheric warming uniformly heats the Dead Sea surface water and, consequently, increases SST. In turn, the increased SST leads to an increase in evaporation, contributing to the steady shrinking of the Dead Sea. However, the uniform heating of Dead Sea surface water by the above mentioned atmospheric warming cannot explain the non-uniform 17-

year mean SST distribution in the summer months, observed by MODIS: this non-uniformity is characterized by maximal SST observed near the coastline, while minimal SST was observed in the middle of the Dead Sea (a new Figure 12 in the revised version). This non-uniform SST distribution demonstrates that the strongest heating of the surface water takes place on the periphery of the lake and not in the middle: this indicates the presence of heat flow from land to sea. Thus, the uniform heating of Dead Sea surface water by the regional atmospheric warming cannot explain such non-uniformity in daytime SST.

Moreover, shown in a new Figure 13, a comparison of the spatial distribution of daytime Dead Sea surface temperature (averaged over the summer months) between the two years: 2000 and 2016 illustrates changes in SST during the study period. We can see that the most significant increase in SST was observed over the north-west and southern sides of the Dead Sea (Fig. 13), where shrinking of the Dead Sea water area was pronounced (Fig. 14b). This fact of the non-uniform heating of Dead Sea surface water cannot be explained by the uniform atmospheric warming over the Dead Sea valley.

Furthermore, such non-uniform heating of Dead Sea surface water in the summer months was characterized by the non-uniform spatial distribution of long-term SST trends during the study period (Fig. 14a). Maximal SST trends of over 0.8 $^o$C decade$^{-1}$ were observed over the north-west and southern sides of the Dead Sea, where shrinking of the Dead Sea water area was pronounced (Figs. 14 a and b). No noticeable SST trends were observed over the eastern side of the lake, where shrinking of the Dead Sea water area was insignificant (Figs. 14 a and b). Thus, satellite-based SST measurements showed correspondence between the location of maximal SST trends and that of Dead Sea shrinking. This correspondence confirms the existence of the positive feedback loop between the shrinking of the Dead Sea and the positive SST trends. This positive feedback loop is a causal factor of the observed non-uniform spatial distribution of long-term SST trends (see a new Section 4 in the revised version).

**Specific comments by Reviewer #2**

1) **C:** *However, a number of factors need to be considered:*
   *1) warming of waters produces expansion and the water level should increase. However, this effect is much more evident in big oceans than in shallow inland water bodies that are subjected to marked year to year changes. A warming atmosphere produces more evaporation, meaning more water is available for precipitation.*

   **A**: According to the Reviewer's remark, "warming of waters produces expansion and the water level should increase". This process might exist, however it could not prevent the observed acceleration in the long-term Dead Sea water-level drop (a new Figure 11b). As for the effect of atmospheric warming, positive trends have been detected in air temperature over Israel over several past decades (Yosef et al., 2018). In a new Section 4 in the revised version, we focused on the summer months when precipitation does not occur. The observed acceleration in the long-term Dead Sea water-level drop in the summer months could not be caused by changes in precipitation (Fig. 11b).

2) **C:** *If the level drops, the density and saltiness might rising until a certain point where the rate of evaporation will reach a kind of equilibrium.*

   **A**: In the summer months, in the absence of precipitation and the insignificant water inflow from the Jordan River, an acceleration in the long-term Dead Sea water-level drop was observed corresponding to the steadily increasing evaporation (a new Figure 11b). Thus, during the study period, there was no equilibrium in the rate of evaporation. The increase in evaporation from year to year is caused by the steady warming of Dead Sea surface water (as a result

of Dead Sea shrinking) and by a positive feedback loop between the shrinking of the Dead Sea and positive SST trends. In addition, the observed atmospheric warming heats the Dead Sea surface water causing an increase in evaporation. Thus, during the study period, in the Dead Sea, the steadily increasing heat flow from land to sea together with atmospheric warming is a causal factor of additional evaporation. This indicates that, during the study period, the density and saltiness of the Dead Sea surface water did not reach the point where the Dead Sea evaporation reached equilibrium.

3) **C:** *The evaporation of water depends on open water surface. For instance the water surface area influences the quantity lost through evaporation from water (if the water surface decreases less surface is exposed to atmosphere). The authors look at one single point that is not enough to draw conclusions about wind effects. This point needs to be better supported (e.g. measuring the water surface area chance over time).*

**A**: The positive SST trends, based on satellite measurements, were observed over the whole area of the Dead Sea (Fig. 1) and not only over one single point. With respect to the Reviewer's remark about the use of wind measurements from only one measuring site, monthly data of wind speed from two other meteorological stations (located in the vicinity of the Dead Sea) were analyzed (a new Figure 6 in the revised version). We refer the Reviewer to our answer to his specific comment 5.

4) **C:** *Evaporation is influenced by the temperature of air above (higher air temperatures favorite the rate of evaporation). But also the vapor pressure of the liquid has to be considered. The rate of evaporation therefore depends on the difference between saturation vapor pressure at the water temperature and at the dew point of the air. Higher the difference, more the evaporation.*

**A**: Over a limited area, such as the Dead Sea valley, the effect of air temperature is uniform. By contrast, as discussed in Section 4 in the revised version, satellite-based SST measurements showed the non-uniform 17-year mean SST distribution in the summer months: this non-uniformity is characterized by maximal SST observed near the coastline, while minimal SST was observed in the middle of the Dead Sea (a new Figure 12 in the revised version). Moreover, satellite-based SST measurements showed correspondence between the location of maximal SST trends and that of Dead Sea shrinking (Figs. 14 a and b). Therefore, the steady shrinking of the Dead Sea causes an increase in SST and, consequently, increases evaporation from the Dead Sea: this leads to the observed accelerating rate of Dead Sea water-level drop (Fig. 11b)

5) **C:** *Another comment is about the wind. More than amplitude, evaporation is influenced by surface roughness. Moreover, nothing is said about air pressure. I would expect decreasing evaporation with increasing pressure.*

**A**: In our study, we analyzed trends in near surface winds. In accordance with the Reviewer's remark 3, in addition to wind measurements taken at the hydrometeorological buoy anchored in the Dead Sea, we analyzed monthly data of near surface wind speed from two other meteorological stations, located in the vicinity of the Dead Sea: Sdom (31.03N, 35.39E) and Ein-Gedi-SPA (31.42N; 35.38E) (see Section 3 in the revised version). There were no statistically significant trends in wind speed taken at the two aforementioned monitoring sites (Fig. 6).
As for the Reviewer's remark about air pressure: Hect and Gertman (2003) discussed a statistically significant positive trend of 1.1 hPa/decade in air pressure, based on their measurements at the hydrometeorological buoy anchored at the Dead Sea, during the 10-year period from 1992 – 2002. They explained this positive trend in air

pressure by the fact that the air pressure gauge mounted on the buoy dropped by approximately 7 m, as a result of the Dead Sea water-level drop during that period, (Hect and Gertman, 2003). According to the Reviewer's remark, he "would expect decreasing evaporation with increasing pressure". This process might exist, however it did not prevent the acceleration in the long-term Dead Sea water-level drop observed during that period of 1992 – 2002 (Fig. 7b in the revised version).
.

6) **C:** *Another point is that most of the water volume of the Jordan River is extracted before the river reaches the Dead Sea. How much is contributing to the lake drop? It is important to know before saying evaporation is main contribution to the decrease of the water level.*

**A**: We consider that, in the summer months, evaporation is the main contributor to the observed decrease in the Dead Sea water level. In every summer month (July, August, September), on average, Dead Sea water level dropped by approximately 0.15 m (a new Figure 10). During the study period of 2000 – 2016, the square of the Dead Sea water area is approximately 600 - 640 $km^2$ (El-Hallaq and Habboub, 2014). Consequently, every summer month the Dead Sea loses approximately 100 x $10^6$ $m^3$ of water.

As for the Jordan River: after the construction of water supply projects in Israel (1964), Jordan (1966) and Syria (1970), the main flow of water into the Jordan River from the Sea of Galilee and from the Yarmouk River was blocked (Holtzman et al., 2005). Since that time, the only flow of fresh surface water into the Jordan River has included rare flood events and negligible contributions from small springs. According to Gidon Bromberg (Yale Environment, #360, 2008, https://e360.yale.edu/features/will_the_jordan_river_keep_on_flowing ): in 2008 he mentioned that "massive water withdrawals for irrigation had created lush areas in the Jordan valley but have reduced the river to a trickle in many spots" (see a new Figure 8 in the revised version). During the period of 2000 – 2001, Holtzman et al. (2005) measured the Jordan River flow rate of approximately 0.5 - 1 $m^3$ $s^{-1}$. Therefore, in every summer month, the water inflow from the Jordan River to the Dead Sea was approximately 1 - 2 x $10^6$ $m^3$. This is less than 2% of the above mentioned amount of Dead Sea water loss per month.

**C:** *To conclude, the topic is certainly interesting, but authors are missing parts that could influence their conclusions. In my opinion, they have to better describe the occurring meteorological, hydrological, and geophysical processes and their interactions with the related variables that could play a key role. Also, the bibliography is just mentioned and somewhat missing (see below where there is important discussion about solar radiation and wind effects) and should be better explained how some conclusions are stated about water level drops, evaporation, winds, etc.*

*e.g. Lensky, N. G., Lensky, I. M., Peretz, A., Gertman, I., Tanny, J., & Assouline, S. (2018). Diurnal course of evaporation from the Dead Sea in summer: A distinct double peak induced by solar radiation and night sea breeze. Water Resources Research, 54(1), 150-160.*

*Overall, the paper calls for some major revision. I like reading the revised version.*

**A**: Various meteorological, hydrological, and geophysical processes could influence Dead Sea evaporation (Lensky et al., 2005, 2018; Metzger et al., 2018). To explain the opposite trends based on available measurements (i.e. the positive SST trends and the accelerating rate of the Dead Sea water-level drop due to a long-term increase in evaporation), our study focuses on the total result of all these factors. We found that, during the study period, the total result of all these factors is a long-term increase in evaporation, particularly in the summer months.

The reference to Lensky et al. (2018) has been added to the Introduction section in the revised version, in accordance with the Reviewer's remark.

We thank Reviewer #2 for his comments on our manuscript, which helps us to clarify the paper.

References

Hecht, A., Gertman, I. (2003). Dead Sea meteorological climate, in Nevo, E., Oren, A., Wasser, S.P. (Eds.) Fungal Life in the Dead Sea. Oberreifenberg, Germany, ISBN 978-3-906166-10-0, pp. 68 - 114.

[revised manuscript text omitted]

---

## Author Response (AR2)

Reply to the Editor

Dear Dr. Katz,

Below, we present our answers (A) to your comments (C):

C1: *Please follow the suggestion of the Reviewer and see if you can improve the summarizing of the scenario. Reviewer #2 wrote "I appreciate the efforts of the authors to improve the original manuscript. I only suggest authors to better summarise the scenario, in particular explaining to the reader why some meteorological, hydrological, and geophysical processes and their interactions with the related variables are not influencing the conclusions."*

A1: According to the suggestion of Reviewer #2, the following sentence has been added to the Abstract and the Conclusions: Our findings of the existence of a positive feedback loop between the positive SST trends and the shrinking of the Dead Sea imply the following significant point: any meteorological, hydrological or geophysical process causing the steady shrinking of the Dead Sea will contribute to positive trends in SST. (Page 2, Lines 26 – 29; Page 10, Lines 4 – 6)

In addition, the following sentences have been added to Section 4 "SST trends and Dead Sea level drops in the summer months": Thus, satellite-based SST measurements showed correspondence between the location of maximal SST trends and that of Dead Sea shrinking: this indicates a causal link between them. This fact implies the following point: any meteorological, hydrological or geophysical process causing the steady shrinking of the Dead Sea also contributes to the positive trends 20 in SST. This is in accordance with the existing positive feedback loop between the positive SST trends and the shrinking of the Dead Sea, as discussed in Section 3. (Page 8, Lines 31 – 33, and Page 9, Lines 1 – 3)

C2: *I feel that Figure 8 is not needed and beyond the scope of the manuscript. Consider omitting it.*

A2: In accordance with the Editor's remark, Figure 8 has been removed from the manuscript.

C3: *P. 8, Lines 13-14: 'over several past decades' appears twice.*

A3: The text has been updated according to the Editor's comment.

[revised manuscript text omitted]